# Calcineurin phosphatase activity regulates Varicella-Zoster Virus induced cell-cell fusion

**Momei Zhou**[1]*, **Vivek Kamarshi**[1], **Ann M. Arvin**[1,2ʘ], **Stefan L. Oliver**[1ʘ]

**1** Department of Pediatrics, Stanford University School of Medicine, Stanford, California, United States of America, **2** Department of Microbiology & Immunology, Stanford University School of Medicine, Stanford, California, United States of America

ʘ These authors contributed equally to this work.
* mzhou6@stanford.edu

## Abstract

Cell-cell fusion (abbreviated as cell fusion) is a characteristic pathology of medically important viruses, including varicella-zoster virus (VZV), the causative agent of chickenpox and shingles. Cell fusion is mediated by a complex of VZV glycoproteins, gB and gH-gL, and must be tightly regulated to enable skin pathogenesis based on studies with gB and gH hyperfusogenic VZV mutants. Although the function of gB and gH-gL in the regulation of cell fusion has been explored, whether host factors are directly involved in this regulation process is unknown. Here, we discovered host factors that modulated VZV gB/gH-gL mediated cell fusion via high-throughput screening of bioactive compounds with known cellular targets. Two structurally related non-antibiotic macrolides, tacrolimus and pimecrolimus, both significantly increased VZV gB/gH-gL mediated cell fusion. These compounds form a drug-protein complex with FKBP1A, which binds to calcineurin and specifically inhibits calcineurin phosphatase activity. Inhibition of calcineurin phosphatase activity also enhanced both herpes simplex virus-1 fusion complex and syncytin-1 mediated cell fusion, indicating a broad role of calcineurin in modulating this process. To characterize the role of calcineurin phosphatase activity in VZV gB/gH-gL mediated fusion, a series of biochemical, biological and infectivity assays was performed. Pimecrolimus-induced, enhanced cell fusion was significantly reduced by shRNA knockdown of FKBP1A, further supporting the role of calcineurin phosphatase activity in fusion regulation. Importantly, inhibition of calcineurin phosphatase activity during VZV infection caused exaggerated syncytia formation and suppressed virus propagation, which was consistent with the previously reported phenotypes of gB and gH hyperfusogenic VZV mutants. Seven host cell proteins that remained uniquely phosphorylated when calcineurin phosphatase activity was inhibited were identified as potential downstream factors involved in fusion regulation. These findings demonstrate that calcineurin is a critical host cell factor pivotal in the regulation of VZV induced cell fusion, which is essential for VZV pathogenesis.

**Data Availability Statement:** The mass spectrometry phosphoproteomics data is available at ProteomeXchange Consortium via the PRIDE partner repository with the dataset identifier PXD018290 and 10.6019/PXD018290. The

Uniphosphoprotein-finder code for phosphoproteomics analysis has been submitted to GitHub at https://github.com/Hub-mz/Uniphosphoprotein-finder.

**Funding:** Research reported here was supported by grants to AMA from National Institutes of Health (NIH), United States, R01 AI102546 and R01 AI20459. The Zr-IMAC phosphopeptide enrichment mass spectrometry utilized the Stanford Cancer Institute Proteomics/Mass Spectrometry Shared Resource which is supported in part by NIH P30 CA124435. The funders had no role in study design, data collection and analysis, decision to publish, or preparation of the manuscript.

**Competing interests:** The authors have declared that no competing interests exist.

## Author summary

VZV is a medically important human virus that causes varicella (chicken pox) and reactivates as zoster (shingles). Postherpetic neuralgia (PHN) is a common sequela of zoster, a persistent nerve pain that can last from months to years after zoster. Refractory to antiviral drugs and pain treatment, PHN significantly impacts quality of life. Despite the success of VZV live-attenuated vaccines in the healthy population, immunocompromised individuals are still vulnerable to significant morbidity. The recently approved subunit zoster vaccine is effective but whether it works against varicella is not known. Therefore, a better understanding of the molecular basis of VZV pathogenesis would provide significant advances towards alternative treatments. A hallmark of VZV pathogenesis is syncytia formation observed in infected skin and nerve ganglia. VZV-induced cell fusion is mediated by the viral fusion complex comprised of glycoproteins gB, gH, and gL. Dysregulation of this fusion process leads to impaired VZV infection in skin. Here, we report that inhibition of a cellular phosphatase, calcineurin, significantly increased gB/gH-gL mediated cell fusion, and intensified syncytia formation during VZV replication, but suppressed virus spread. Our study demonstrated that calcineurin phosphatase activity regulates VZV-induced cell fusion, providing a new perspective for potential antiviral strategies that target host factors.

## Introduction

Glycoproteins in the membranes of enveloped viruses mediate host entry by inducing fusion between viral and cellular membranes. Importantly, these fusogenic glycoproteins can also induce cell fusion, which is recognized as a component of pathogenesis in the infected host for viruses from diverse families. Among human pathogens, these viruses include the paramyxoviruses, measles virus (MV), respiratory syncytial virus (RSV), human immunodeficiency virus (HIV), and the herpesviruses, varicella-zoster virus (VZV) and herpes simplex virus (HSV) [1–3]. While differentiated host cells rarely fuse, except macrophages or during bone, muscle, and placental development and tissue regeneration from stem cells [4], these viruses overcome the intrinsic barrier of cell fusion by utilizing a fusogenic glycoprotein or fusion complexes to reduce the repulsive energy between two hydrophobic membranes and bring them into close proximity to form the initial fusion pore [5]. Expansion of the fusion pore leads to complete fusion of plasma membranes, cytoplasmic content mixing and the formation of multinucleated syncytia. Although functions have been assigned to virally encoded fusogens, identifying host cell factors involved in the disruption of cell homeostasis that accompanies virus-induced cell fusion is a largely unexplored concept central for understanding the mechanisms of viral pathogenesis. Here, we investigate host cell factors involved in VZV induced cell fusion because this process is a hallmark of VZV pathogenesis and is linked directly to the adverse health consequences of VZV infection [6].

VZV causes varicella (chickenpox) during primary infection, and zoster (shingles) upon reactivation from latently infected ganglionic neurons. VZV reactivation can lead to significant health complications, including the debilitating condition of postherpetic neuralgia (PHN) and cerebral artery vasculitis, a precursor of stroke [7,8]. Although live attenuated VZV vaccines for varicella reduce disease burden, they are unsafe for immunodeficient patients [9,10]. A recombinant glycoprotein E (gE) vaccine is effective against zoster but its efficacy against varicella is unknown [11]. Antiviral drugs can control VZV infection but do not alleviate pain associated with PHN [12]. Therefore, a better understanding of the

molecular mechanisms of VZV pathogenesis is needed to derive interventions that overcome these limitations.

VZV cell fusion leads to the characteristic polykaryocytes formation within varicella and zoster skin lesions [2,6,13]. Syncytia formation in melanoma (MeWo) cells infected with VZV simulates this remodeling of cells in the skin tissue microenvironment. Importantly, fusion can occur between neurons and satellite cells in sensory ganglia during VZV reactivation, which might have a role in PHN. VZV infection of human skin and dorsal root ganglia xenografts in the severe combined immunodeficiency (SCID) mouse model recapitulates these cell fusion events in vivo [14–18].

All herpesviruses share a conserved core fusion machinery comprised of glycoprotein B (gB) and the heterodimer gH-gL; gB is the primary fusogen while gH-gL is required to prime the fusion reaction [19]. The ectodomains of VZV gB and gH have residues required for cell fusion as shown by mutagenesis [17,18,20–23]. Ectodomain mutations that reduce or abolish cell fusion can inactivate VZV or impair pathogenesis in skin xenografts [18,21,22]. Conversely, tight control of fusion is equally important for VZV pathogenesis and is mediated by the cytoplasmic domains (CTDs) of gB and gH [18,24–26]. A virus-free cell fusion assay developed for VZV demonstrated that gB, gH and gL are necessary and sufficient to drive cell fusion when the last eight amino acids of the gH C-terminus (gH[TL]) are deleted [21,22,25,27,28]. The VZV fusion assay has been a powerful tool utilized to identify functional domains in the gB/gH-gL fusion complex [18,22,23,25,26,28,29]. In contrast to VZV, HSV also requires gD in addition to the core complex of fusion proteins [30,31]. Unlike HSV, the length of the VZV gH CTD is important for regulating fusion as shown by exaggerated syncytia formation when the gH[TL] mutation is introduced into the virus, and suggests that this modification facilitates detection of fusion in the gB/gH-gL virus-free cell fusion assay [25,32]. Critically, incorporation of the two gB and gH CTD regulatory domains mutations, gB[Y881F] and gH[TL], into the VZV genome showed that while the gB and gH ectodomain functions were preserved, nucleocapsid production and viral particle assembly were prevented, demonstrating that VZV induced cell fusion can be decoupled from virus spread [25]. In addition, analyses of several non-inactivating, hyperfusogenic gB and gH CTD mutant viruses link impaired VZV propagation to the accelerated sequestration of many nuclei within syncytia and disruption of the actin cytoskeleton, resulting in a cellular environment that suppresses virus spread and, counterintuitively, yields smaller plaques [18,25,26,33].

Notably, a differential cellular gene expression profile was associated with infection by the hyperfusogenic gB[Y881F] CTD mutant virus compared to intact virus, suggesting a role for cell proteins in the fusion regulation [33]. Here, we report host cell factors implicated in the regulation of VZV gB/gH-gL mediated cell fusion using a high-throughput stable reporter fusion assay (HT-SRFA) that evaluates the effects of bioactive compounds on cell fusion. This approach identified a subset of non-antibiotic macrolides that dramatically increased gB/gH-gL mediated cell fusion by inhibiting the phosphatase activity of calcineurin. This effect was not limited to VZV but had broad activity as quantified using the HSV-1 fusion complex and syncytin-1, a host cell retrovirus-derived fusogen. Moreover, calcineurin phosphatase activity was maintained in VZV infected cells and its inhibition enhanced syncytia formation during VZV replication but suppressed virus spread. Calcineurin inhibition was associated with distinctive phosphorylation of seven cell proteins, which might be downstream host cell factors. These findings implicate calcineurin as a host cell factor that functions together with the gB/gH-gL complex to ensure that VZV induced cell fusion is effectively regulated as required to sustain VZV pathogenesis.

## Results

### Identification of compounds that affected VZV gB/gH-gL mediated cell fusion using a HT-SRFA

To identify cellular factors that regulate VZV cell fusion, a HT-SRFA was developed to assess the differential effects of well-defined bioactive compounds on VZV gB/gH-gL mediated cell fusion between effector CHO cells expressing the glycoproteins and target MeWo cells (Fig 1A). The gH[TL] construct lacking the last eight amino acids in the CTD was selected for the HT-SRFA because it provides a superior signal-to-noise ratio for the fusion assay without affecting the fusion functions of the gH ectodomain [28]. Libraries comprising 4,846

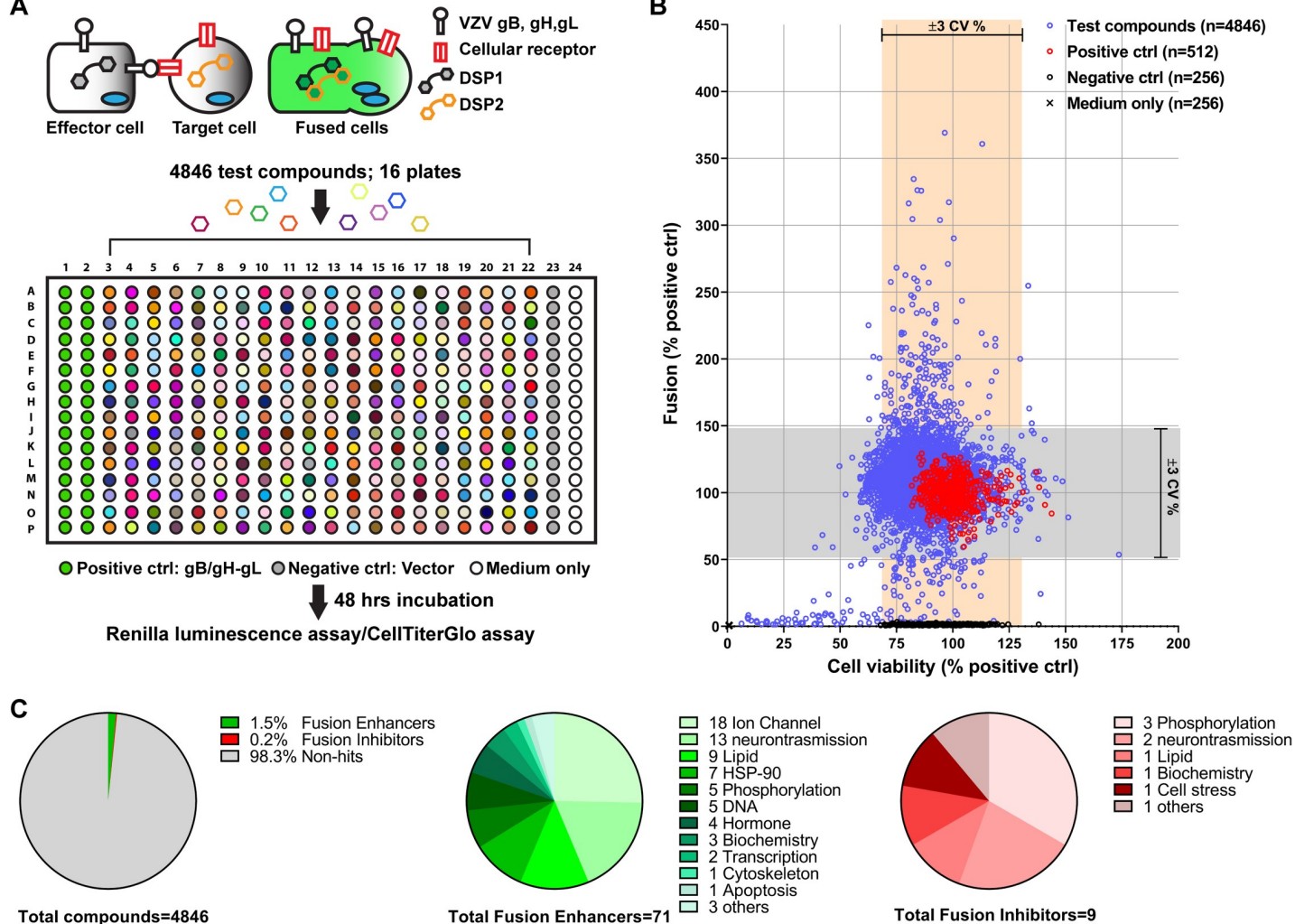

**Fig 1. Bioactive compounds that affect VZV gB/gH-gL-mediated cell fusion identified using a HT-SRFA.** (A) Schematic of the HT-SRFA with *Renilla* luciferase dual split protein (DSP). Fusion of effector cells, CHO-DSP1, transiently expressing VZV glycoproteins, gB and gH[TL]-gL, with target cells, MeWo-DSP2, reconstitutes the *Renilla* luciferase DSP. Effector and target cells were seeded into 384-well plates, and treated with compound libraries for 48 hrs before measuring fusion efficiency; cell viability was assessed by CellTiter-Glo. Positive controls were wells without drug; negative controls had effector cells transfected with empty vectors; medium only controls were included for cell viability. (B) Scatter plot of cell fusion and cell viability derived from the HT-SRFA. The Y-axis and X-axis indicate fusion efficiency and cell viability values normalized to the mean of positive controls. The mean of the percentage (% positive ctrl) from two biological replicates are shown: blue circles are all 4,846 compounds screened, red and black circles are positive and negative controls and the black crosses are medium only. Grey and orange boxes show ±3 CV % of positive controls for fusion efficiency and cell viability respectively. (C) Frequency (%) of compounds that affect cell fusion identified by the HT-SRFA (left), compound classes of fusion enhancers (center) and inhibitors (right), and number of compounds in each class.

compounds from the NIH clinical collection (NIHCC), the Library of Pharmacologically Active Compounds (LOPAC), Microsource Spectrum, ICCB known Bioactives, and FDA approved drugs were selected because these compounds have known cellular targets. After elimination of compounds that were cytotoxic or were included more than once across the libraries, 80 unique compounds (1.7%) were found to significantly increase (71 compounds; 1.5%) or decrease (9 compounds; 0.2%) gB/gH-gL dependent cell fusion based on a ±3 coefficient of variation (CV) compared to the mean of untreated positive controls ($\geq$ 144.7% or $\leq$ 55.3%) (**Fig 1B and 1C** and **S1 Table**). The 80 compounds fell into pharmacological drug classes (LOPAC library classification) involved in relatively few but diverse categories of biological activity. The most common categories of cell targets for drugs that enhanced fusion included ion channels and lipid biogenesis, while those targeted by inhibitors of fusion were most often components of phospho-protein signaling pathways (**Fig 1C**). Of interest, given the neurotropism of VZV, compounds that target proteins involved in neurotransmission were found to either enhance or inhibit gB/gH-gL mediated cell fusion. The fact that only 1.7% of compounds in the library altered gB/gH-gL mediated cell fusion and that many were in the same drug class supported the capacity of the HT-SRFA to reveal a subset of cellular factors potentially involved in VZV induced cell fusion.

The robustness of the HT-SRFA was corroborated by the finding that different compounds with similar cellular targets had consistent effects on fusion; for example, six different calcium channel inhibitors all increased fusion, implicating calcium signaling pathways in the regulation of fusion (**S1 Table**). Tacrolimus (FK506), a non-antibiotic macrolide that also has effects linked to calcium signaling, enhanced fusion to 163.9% of the untreated positive controls at 8.33 μM in the HT-SRFA (**S1 Table**). Tacrolimus, which is known for suppression of T cell function to prevent organ transplant rejection, functions by binding the FK506-binding protein 12 (also known as FKBP1A) and directing the drug-bound FKBP1A to bind to calcineurin, which then specifically inhibits the phosphatase activity of this cell protein [34–36]. Calcineurin, also known as protein phosphatase 3, is a calcium-dependent serine/threonine phosphatase, which can be activated by a conformational change in response to elevated intracellular calcium levels. The binding of the tacrolimus/FKBP1A complex to calcineurin inhibits its phosphatase activity likely by physically blocking the access of macromolecular substrates of calcineurin to the phosphatase active site [34,35]. The inhibitory effect of tacrolimus on calcineurin would be expected to be analogous to the consequences of reduced intracellular calcium levels on this cellular phosphatase due to calcium channel inhibition. Thus, the increase in fusion by tacrolimus was consistent with enhanced fusion by calcium channel inhibitors. Therefore, tacrolimus and related compounds were selected to investigate whether calcineurin was involved in the regulation of VZV gB/gH-gL mediated fusion.

## Inhibition of calcineurin phosphatase activity enhanced VZV gB/gH-gL mediated cell fusion

Tacrolimus, pimecrolimus and sirolimus (also known as rapamycin) are closely related non-antibiotic macrolides (**Fig 2A**). Like tacrolimus, pimecrolimus also binds to FKBP1A and the binary complex specifically inhibits the phosphatase activity of calcineurin, whereas sirolimus binds to FKBP1A and inhibits mTOR but not calcineurin [37,38] (**Fig 2B**). Consistent with their similar mechanisms of action, the chemical structures of tacrolimus and pimecrolimus differ only by replacement of a hydroxyl group with chloride and conversion of an ethyl to a methyl group. Sirolimus shares the FKBP1A-binding site with tacrolimus and pimecrolimus but differs primarily by having an interface that interacts with mTOR rather than calcineurin [39,40] (**Fig 2A**). Thus, comparing the effects of these drugs allowed an evaluation of the role of calcineurin phosphatase activity in VZV gB/gH-gL mediated cell fusion.

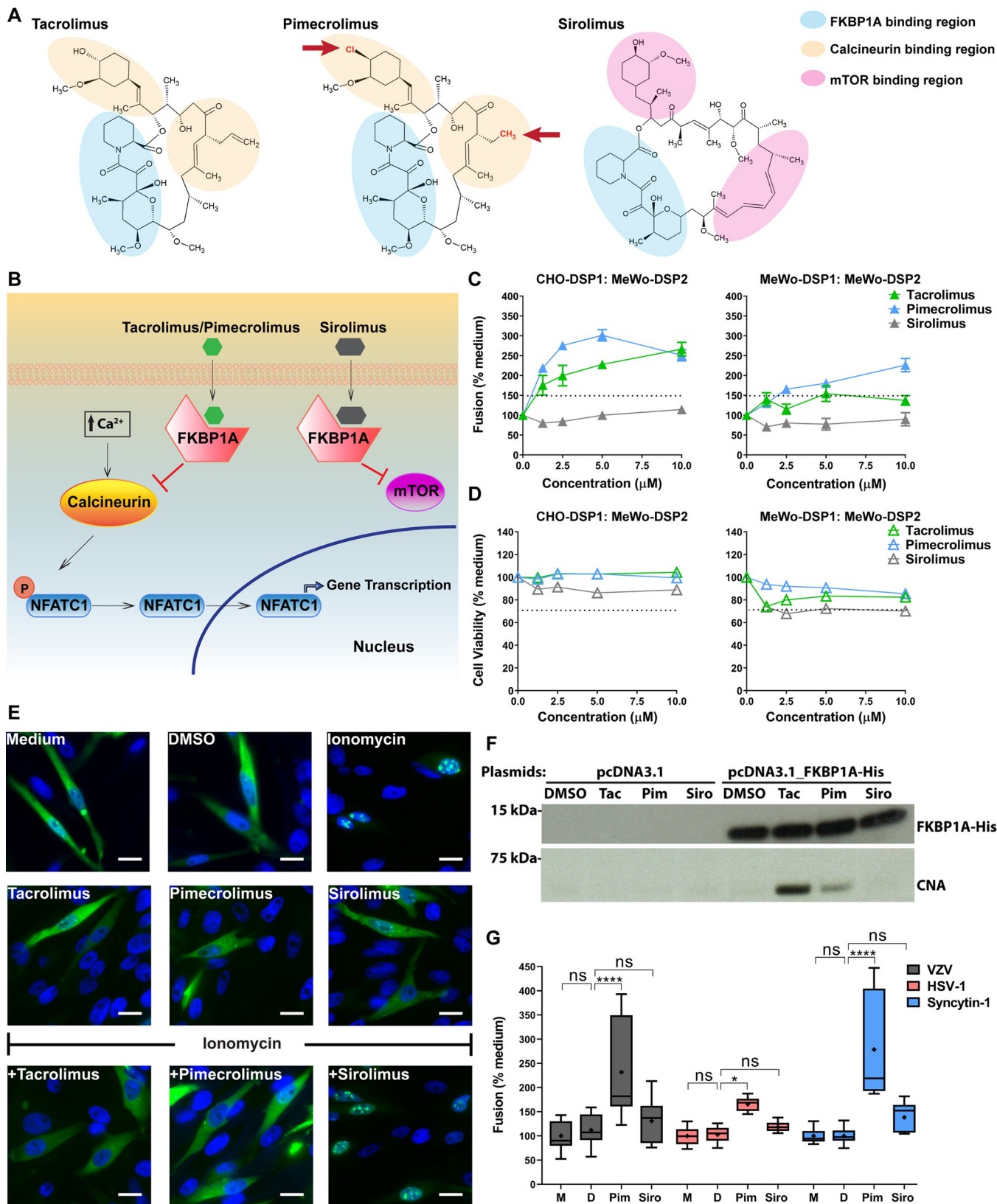

**Fig 2. Inhibition of calcineurin phosphatase activity enhances VZV gB/gH-gL mediated cell fusion.** (A) Chemical structures of tacrolimus, pimecrolimus and sirolimus; pimecrolimus group substitutions indicated by red arrows. Regions of interaction with FKBP1A (blue), calcineurin (yellow) and mTOR (red).

(B) Schematic of drug interactions with cellular factors. (C and D) Cell fusion and cell viability dose-response curves to tacrolimus, pimecrolimus and sirolimus. CHO-DSP1 or MeWo-DSP1 cells transiently expressing VZV gB/gH[TL]-gL co-cultured with MeWo-DSP2 cells, treated with drug at indicated concentrations. Cell fusion efficiency (C) and cell viability (D) were quantified and normalized to positive controls (medium; no drug). Data are represented as mean ± standard error of the mean (SEM) for ≥3 independent experiments. Dash lines indicate the cutoff for statistically significant enhanced fusion or cytotoxicity. (E) Fluorescence microscopy of GFP-NFATC1 nuclear translocation to demonstrate calcineurin phosphatase activity in MeWo cells. GFP-NFATC1 nuclear translocation induced by ionomycin-triggered calcineurin activation (ionomycin; upper right panel) in MeWo cells and prevention by treatment with tacrolimus (+Tacrolimus) and pimecrolimus (+Pimecrolimus). Nuclei stained with Hoechst 33342 (blue) and GFP-NFATC1 (green). Representative fluorescence microscopy images are shown from three independent experiments. Scale bars = 15 μm. (F) Tacrolimus and pimecrolimus induced binding of FKBP1A and calcineurin in MeWo cells. Western blots of FKBP1A-His (anti-FKBP1A) and calcineurin (anti-calcineurin subunit A; CNA) in eluates from CHO cells transfected with His-tagged FKBP1A or control plasmids that were lysed, precipitated with nickel agarose beads, then mixed with MeWo cell extract and treated with DMSO, tacrolimus (Tac; 10 μM), pimecrolimus (Pim; 10 μM) or sirolimus (Siro; 10 μM). (G) Box and whisker plots for cell fusion quantified by the SRFA using CHO-DSP1 cells transfected with plasmids expressing either VZV gB/gH[TL]-gL, HSV-1 gB/gH-gL/gD, or syncytin-1 and mixed with MeWo-DSP2, untreated (medium; M) or treated with DMSO (D), pimecrolimus (Pim; 10 μM) or sirolimus (Siro; 10 μM) for 48 hrs. Fusion efficiency was measured and normalized to medium (% medium). Boxes represent 25–75 percentile, whiskers extend to 10–90 percentile, the median is the horizontal band, and the mean (+) is from three independent experiments. Statistical differences were evaluated by two-way ANOVA (ns, not significant; $^*$, $p < 0.05$; $^{****}$, $p < 0.0001$).

In dose response experiments, none of the drugs elicited cell fusion in the absence of the VZV fusion complex gB and gH-gL (S1 Fig), confirming that cell fusion was driven by the VZV glycoproteins and indicating that the compounds affected the cellular conditions involved in the regulation of the viral fusion process. As expected, tacrolimus significantly enhanced fusion in a dose-responsive fashion, with increases to 228.1% (5 μM) and 266.6% (10 μM). Pimecrolimus also dramatically enhanced fusion to 301.5% (5 μM) and 250.9% (10 μM) whereas sirolimus remained inactive at all concentrations (Fig 2C). The HT-SRFA is based on the detection of fusion between CHO cells (CHO-DSP1) expressing gB/gH-gL and target melanoma cells (MeWo-DSP2). However, CHO cells do not support active VZV infection whereas MeWo cells are permissive for VZV infection, which results in cell fusion. To determine whether tacrolimus and pimecrolimus also exert their effects on fusion between MeWo cells, MeWo-DSP1 cells expressing VZV gB and gH-gL were used as effector cells and MeWo-DSP2 cells as target cells. Again, tacrolimus increased fusion (137.3%; 10 μM) and pimecrolimus markedly enhanced fusion (226.4%; 10 μM), but sirolimus had no effect (Fig 2C). Cytotoxicity was not observed with any of the compounds in either system (Fig 2D). Since both tacrolimus and pimecrolimus inhibited calcineurin while sirolimus did not, interference with the phosphatase activity of calcineurin was implicated in the dysregulation of VZV gB/gH-gL mediated cell fusion.

Critically, the enhanced gB/gH-gL mediated fusion by pimecrolimus was not attributable to increased quantities of glycoprotein production because both total and cell surface amounts of gB and gH were lower in transfected CHO-DSP1 cells treated with pimecrolimus compared to medium or DMSO treatment (S2A Fig). Because the fusion was still enhanced by pimecrolimus, these data indicated that the detectable gB and gH quantities were within the range needed for gB/gH-gL mediated fusion. Cell surface amounts correlated with total quantities of gB and gH (S2B Fig), showing that glycoprotein trafficking to the cell surface was not dysregulated by pimecrolimus.

Calcineurin relays signals by dephosphorylating various downstream interaction partners, including the transcription factors of the Nuclear Factor of Activated T cells (NFAT) family [41,42]. To demonstrate that the phosphatase activity of calcineurin was functional in MeWo cells, the well-established fact that activated calcineurin dephosphorylates NFAT and triggers its nuclear translocation was exploited [43]. In response to ionomycin, NFATC1, a member of the NFAT family, has been demonstrated to translocate from the cytoplasm to the nucleus [44], which was used as the basis of a NFATC1 nuclear translocation assay to measure calcineurin phosphatase activity in MeWo cells. As expected, ionomycin (1 μM) resulted in nuclear translocation of GFP-NFATC1, which otherwise was largely diffuse in both the cytoplasm and

nucleus of cells treated with medium or DMSO. None of the drugs influenced GFP-NFATC1 localization alone. However, adding either of the calcineurin inhibitors, tacrolimus or pimecrolimus, prior to ionomycin treatment prevented nuclear translocation of GFP-NFATC1 whereas sirolimus did not (**Fig 2E**). Consistent with this observation, the biochemical interaction between FKBP1A and calcineurin was confirmed in MeWo cells treated with tacrolimus and pimecrolimus (**Fig 2F**). As expected from their failure to alter gB/gH-gL mediated fusion, the FKBP1A-calcineurin complex was not detected in the presence of sirolimus or the DMSO vehicle control. Thus, the compounds that enhanced gB/gH-gL mediated cell fusion specifically triggered the physical FKBP1A-calcineurin interaction, thereby directly inhibiting the phosphatase activity of calcineurin in MeWo cells.

## Inhibition of calcineurin phosphatase activity enhanced HSV-1 gB/gH-gL/gD and syncytin-1 mediated cell fusion

To investigate whether the enhanced cell fusion via interference with calcineurin phosphatase activity was specific to the VZV fusogenic complex, drug effects on other fusogens were evaluated. HSV-1 gB, gH-gL and gD represented the viral fusogen complex from another alphaherpesvirus; in this assay, fusion was detectable using intact gH. Syncytin-1 is a host cell fusion protein encoded by a human endogenous retroviral element, expressed on the surface of placental trophoblasts and implicated in cell-cell fusion during placental development [45,46]. Of note, MeWo cells, which are the target cells in this fusion assay, express the receptors for gD (Herpesvirus entry mediator encoded by TNFRSF14 gene) and syncytin-1 (ASCT2 encoded by SLC1A5 gene) [33]. Pimecrolimus significantly increased fusion induced by both the HSV-1 fusogenic complex and syncytin-1 when compared to DMSO vehicle control (**Fig 2G**). As expected, sirolimus did not affect fusion. This result suggested that the phosphatase activity of calcineurin is part of the host cell fusion regulatory mechanism, irrespective of whether cell fusion is induced by a viral or host fusogen. Without its phosphatase activity, calcineurin regulation of the capacity of these fusogens to modulate the intrinsic barrier against cell fusion is impaired.

## Pimecrolimus disruption of calcineurin-dependent regulation of gB/gH-gL fusion requires FKBP1A

As pimecrolimus binds FKBP1A to relay its inhibitory effect on calcineurin, the knockdown of FKBP1A in MeWo cells was performed to determine whether the enhancing effect of pimecrolimus on VZV gB/gH-gL mediated cell fusion was mitigated. FKBP1A knockdown was validated at both mRNA (83.5% decrease) and protein (86.9% decrease) levels in a stable MeWo-DSP1 cell line expressing shRNA specific for FKBP1A, compared to MeWo-DSP1 control cells (**Fig 3A**). The effect of FKBP1A knockdown on the inhibition of calcineurin phosphatase activity by pimecrolimus was tested using the NFATC1 nuclear translocation assay. At baseline, in both control and FKBP1A knockdown MeWo-DSP1 cells, 5% of GFP-NFATC1 positive cells had NFATC1 nuclear localization, 60% had diffuse cytoplasmic and nuclear expression and 35% had cytoplasmic expression only. Both cell lines responded similarly to ionomycin-induced activation of calcineurin, with GFP-NFATC1 nuclear translocation occurring in 70% of cells. Pretreatment of FKBP1A knockdown cells with pimecrolimus prior to ionomycin produced a 7-fold increase in GFP-NFATC1 nuclear translocation and a 3-fold decrease in cytoplasmic GFP-NFATC1 compared to the control cells (**Fig 3B**). This established that inhibition of calcineurin phosphatase activity by pimecrolimus depended primarily on its interaction with FKBP1A in MeWo cells. Knockdown of FKBP1A marginally reduced gB/gH-gL mediated cell fusion but was not significantly different when compared to MeWo-DSP1

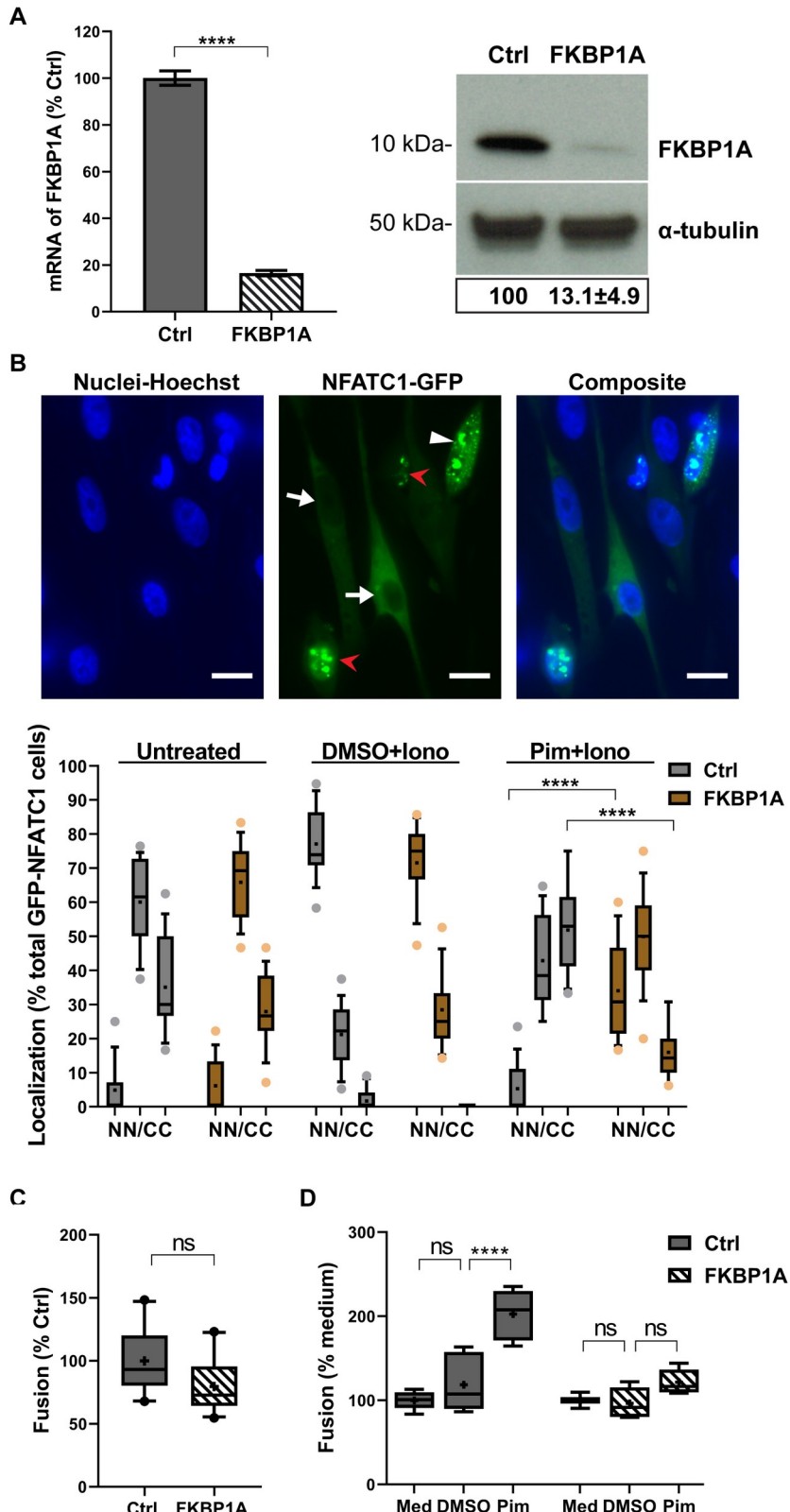

**Fig 3. Pimecrolimus disruption of calcineurin-dependent regulation of gB/gH-gL fusion requires FKBP1A.** (A) FKBP1A shRNA knockdown (KD) in MeWo cells. FKBP1A Transcript levels detected by RT-qPCR in MeWo-DSP1 control cells or FKBP1A KD MeWo-DSP1 cells were normalized to housekeeping gene PGM1 and presented as a

percentage of control cells, with mean ± SEM from three independent experiment (two-tailed, unpaired t-test, ****, $p < 0.0001$). Western blot of FKBP1A protein (anti-FKBP1A). FKBP1A band density was normalized to α-tubulin in control and FKBP1A KD cells respectively. The ratio of the two values was presented as a percentage of control cells. A representative picture, and mean ± SEM of the ratio from three independent experiments are shown. (B) Quantification of calcineurin phosphatase activity in FKBP1A KD cells using the NFATC1 nuclear translocation assay. MeWo-DSP1 control cells or FKBP1A KD MeWo-DSP1 cells transiently expressing GFP-NFATC1 were untreated (medium) or treated with DMSO or pimecrolimus (Pim, 10 μM) 30 min before ionomycin (Iono, 1 μM; 30 min). Fluorescence microscopy images of GFP-NFATC1 (green), nuclei stained with Hoechst 33342 (blue), and a composite image. Scale bars = 15 μm. Red arrowheads indicate nucleus localization (*N*), white arrowhead indicates diffused localization in both nucleus and cytoplasm (*N/C*), and white arrows indicate cytoplasm localization (*C*). Box and whisker plots of GFP-NFATC1 positive cells. The percentage of GFP positive cells in control and FKBP1A KD cells with GFP-NAFTC1 localization to N, N/C, and C. Boxes represent 25–75 percentile, whiskers extend to 10–90 percentile, the median is the horizontal band, and the mean (+) of cells (n = 15 fields of view, 13–22 cell per field of view) from two independent experiments. Dots are outliers. Statistical difference were analyzed by two-way ANOVA for each treatment condition (Untreated, DMSO+Iono, and Pim+Iono) compared to that in the control cells, and significantly different pairs are shown (****, $p < 0.0001$). (C and D) Effect of FKBP1A KD on VZV gB/gH-gL-mediated fusion and response to pimecrolimus. Box and whisker plots for cell fusion quantified by the SRFA in MeWo-DSP1 control cells or FKBP1A KD MeWo-DSP1 cells transfected with plasmids to express VZV gB/gH[TL]-gL and co-cultured with MeWo-DSP2 cells, untreated (medium; Med) or treated with DMSO or pimecrolimus (Pim, 10 μM) for 48 hrs. Cell fusion efficiency when untreated was compared between FKBP1A KD cells and control cells, shown as a percentage of control cells (C), and the response to pimecrolimus was evaluated by normalization to the untreated (% medium) per control and FKBP1A KD cell line respectively (D). Boxes represent 25–75 percentile, whiskers extend to 10–90 percentile, the median is the horizontal band, and the mean (+) from three independent experiments. Dots are outliers. Statistical differences were evaluated by unpaired, nonparametric Mann-Whitney test (C) (ns, not significant) or two-way ANOVA (D) (ns, not significant; ****, $p < 0.0001$).

control cells (**Fig 3C**). As expected, while pimecrolimus significantly increased gB/gH-gL mediated fusion using the MeWo-DSP1 control cells compared to DMSO, the drug failed to induce a significant increase of fusion using the FKBP1A knockdown MeWo-DSP1 cells (**Fig 3D**). These findings corroborated the NFATC1 nuclear translocation data, indicating that FKBP1A was the specific cellular protein bound by pimecrolimus to produce the inhibitory effect on calcineurin and confirmed that dysregulation of VZV gB/gH-gL mediated cell fusion by pimecrolimus was via the inhibition of calcineurin phosphatase activity.

## Calcineurin supports VZV gB/gH-gL-mediated cell fusion

Calcineurin is a heterodimer comprised of catalytic subunit A (CNA) and regulatory subunit B (CNB). The association of CNB to CNA is required for calcineurin activation to enable its phosphatase function [47,48]. Since CNB1 is the only CNB isoform expressed in MeWo cells [33], CNB1 was knocked down by shRNA to directly evaluate calcineurin effects on VZV gB/gH-gL mediated cell fusion. The CNB1 shRNA moderately reduced both mRNA (77.0% decrease) and protein (42.3% decrease) levels of CNB1 (**Fig 4A**). VZV gB/gH-gL induced fusion was significantly reduced in CNB1 knockdown MeWo-DSP1 cells, to only 39.7% of that in MeWo-DSP1 control cells (**Fig 4B**), suggesting that the intact calcineurin heterodimer is important for VZV gB/gH-gL to execute fusion. Pimecrolimus enhanced fusion in CNB1 knockdown MeWo-DSP1 cells when compared to DMSO, but the level of increase was significantly less compared to that observed in MeWo-DSP1 control cells (**Fig 4C**). The detection of residual enhanced fusion by pimecrolimus was likely due to the incomplete knockdown of CNB1 mRNA and the limited reduction in protein production.

## Calcineurin phosphatase activity was maintained in VZV infected cells

To assess the relevance of calcineurin activity for VZV infection, NFATC1 nuclear translocation assays were performed with MeWo cells transiently expressing GFP-NFATC1 that were infected with a VZV recombinant expressing RFP (pOka-TK-RFP) (**Fig 5A**). Without

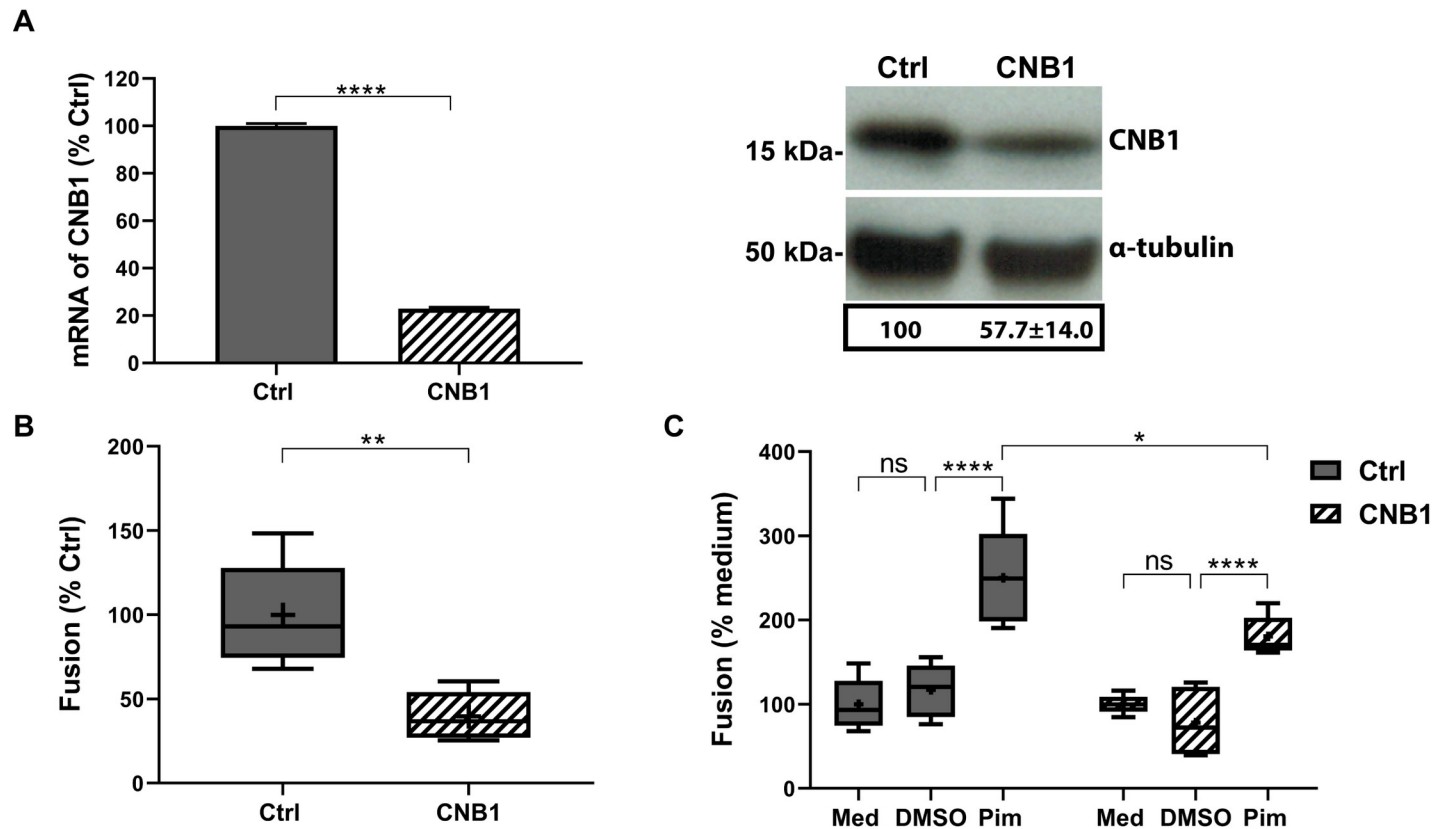

**Fig 4. Calcineurin supports VZV gB/gH-gL mediated cell fusion.** (A) CNB1 shRNA knockdown (KD) in MeWo cells. CNB1 transcript levels in MeWo-DSP1 control cells or CNB1 KD MeWo-DSP1 cells detected by RT-qPCR were normalized to housekeeping gene PGM1 and presented as a percentage of control cells, with mean ± SEM from two independent experiment (two-tailed, unpaired t-test, ****, $P < 0.0001$). Western blot of CNB1 protein (anti-CNB1). CNB1 band density was normalized to α-tubulin in control and CNB1 KD cells respectively. The ratio of the two values was presented as a percentage of control cells. A representative picture, and mean ± SEM of the ratio from two independent experiments are shown. (B and C) Effect of CNB1 KD on VZV gB/gH-gL-mediated fusion and response to pimecrolimus. Box and whisker plots for cell fusion quantified by the SRFA in MeWo-DSP1 control cells or CNB1 KD MeWo-DSP1 cells transfected with plasmids to express VZV gB/gH[TL]-gL and co-cultured with MeWo-DSP2 cells, untreated (medium; Med) or treated with DMSO or pimecrolimus (Pim, 10 μM) for 48 hrs. Cell fusion efficiency when untreated was compared between CNB1 KD cells and control cells, shown as a percentage of control cells (B), and the response to pimecrolimus was evaluated by normalization to the untreated (% medium) per control and CNB1 KD cell line respectively (C). Boxes represent 25–75 percentile, whiskers extend to 10–90 percentile, the median is the horizontal band, and the mean (+) from two independent experiments. Statistical differences were evaluated by unpaired, nonparametric Mann-Whitney test (B) (**, $p < 0.01$) or two-way ANOVA (C) (ns, not significant; *, $p < 0.05$; ****, $p < 0.0001$).

ionomycin treatment, 88% of GFP-NFATC1 was localized in the cytosol of VZV infected cells, while ionomycin treatment triggered a dramatic translocation of 94% of GFP-NFATC1 into the nuclei of infected cells (**Fig 5B**), indicating that calcineurin phosphatase was retained during infection. Importantly, pimecrolimus pretreatment successfully prevented the ionomycin induced nuclear translocation of GFP-NFATC1 (**Fig 5B**), confirming that pimecrolimus directly inhibited calcineurin phosphatase activity in VZV infected cells.

### Inhibition of calcineurin phosphatase activity enhanced cell fusion but suppressed VZV spread

To determine whether calcineurin phosphatase activity was needed to regulate cell fusion during VZV replication, the number of nuclei present within syncytia of Life-Act-tGFP MeWo cells infected with pOka-TK-RFP was quantified in response to treatment with pimecrolimus or the DMSO control (**Fig 6A**). While syncytium expansion was a dynamic movement involving incorporation of newly contacted cells at the periphery of the syncytium during the

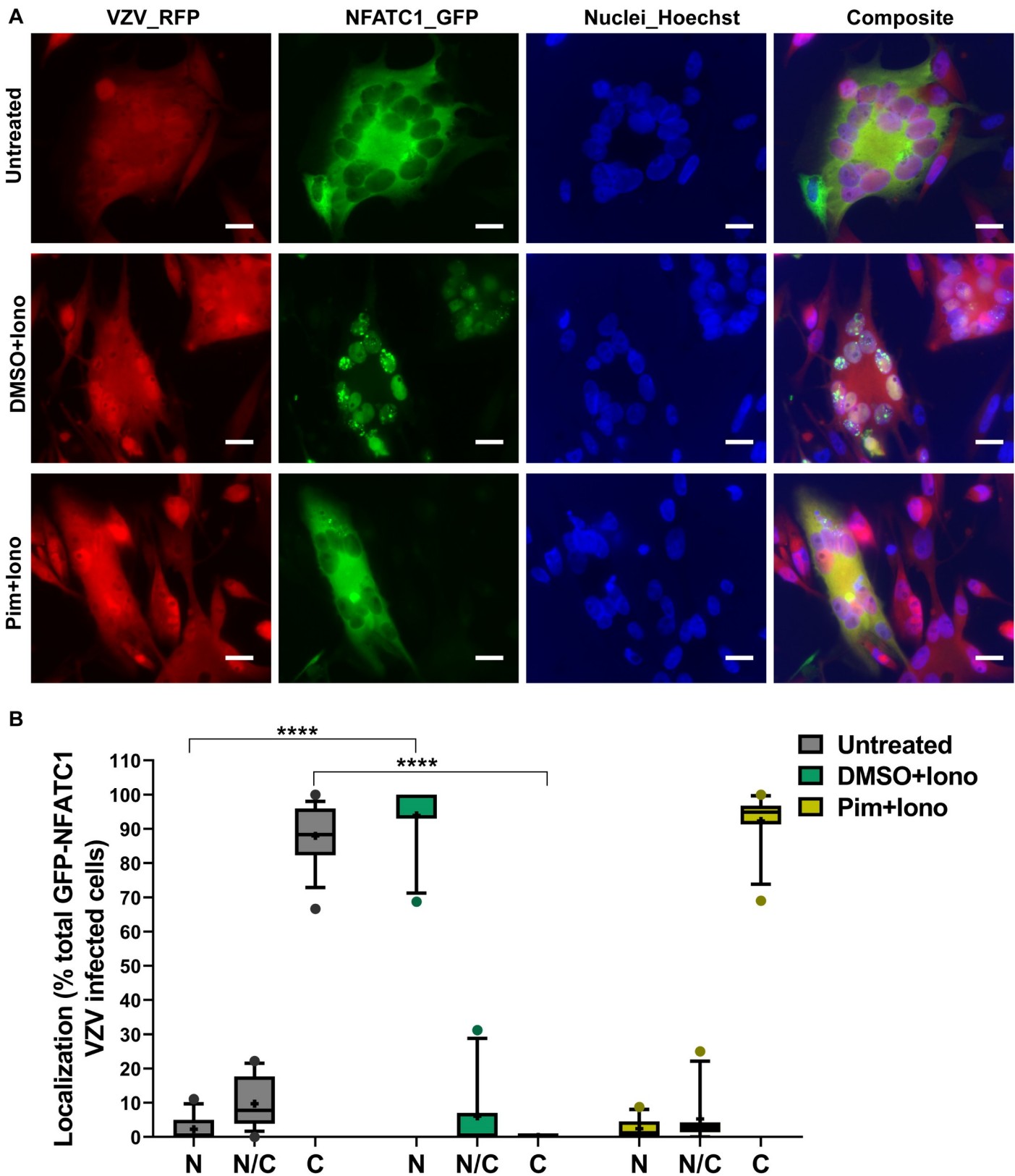

**Fig 5. Calcineurin phosphatase activity remains functional in VZV infected cells.** (A) Fluorescence microscopy images of MeWo cells transiently expressing GFP-NFATC1 at 16 hpi with pOka-TK-RFP that were untreated (medium) or treated with DMSO or pimecrolimus (Pim, 10 μM) 30 min prior to ionomycin

stimulation (Iono, 1 μM; 30 min); pOka-TK-RFP (red), GFP-NFATC1 (green), and nuclei stained with Hoechst 33342 (blue), and a composite image. Scale bars = 15 μm. (B) Box and whisker plots of GFP-NFATC1 localization in VZV infected cells. Of the total GFP-NFATC1 expressing cells that were also infected with TK-RFP, the percentage of cells with GFP-NAFTC1 translocated to nucleus (*N*), diffused in nucleus and cytosol (*N/C*), or localized in cytosol (*C*). Boxes represent 25–75 percentile, whiskers extend to 10–90 percentile, the median is the horizontal band, and the mean (+) of cells (n = 12 fields of view, 41–66 cell per field of view) from three independent experiment. Dots are outliers. Statistical difference were analyzed by two-way ANOVA for each localization group (N, N/C, and C) compared to that in the untreated, shown are the significantly different pairs (****, $p < 0.0001$).

recorded 24 to 40 hours post infection (hpi) (**S1 and S2 Movies**), the time window for a linear increase in nuclei within syncytia was between 24 to 30 hpi (**Fig 6A and 6B**). Fusion was increased in pimecrolimus-treated cells, resulting in significantly more nuclei within syncytia compared to DMSO control between 24 to 30 hpi (**Fig 6C**). The increased fusion of VZV-infected cells when calcineurin phosphatase activity was inhibited corroborated the observations in the gB/gH-gL SRFA in the absence of other VZV proteins. Critically, these data showed that the regulatory role of calcineurin in VZV gB/gH-gL mediated fusion was not due to the use of gH[TL] in the SRFA.

Previous studies have demonstrated that dysregulated cell fusion due to disrupted control by the CTDs of gB and gH results in impaired skin pathogenesis and leads to poor VZV spread, evident from reduced plaque sizes and limited penetration into the MeWo cell monolayer at plaque margins [18,25,26]. To establish whether enhanced cell fusion in response to interference with calcineurin regulation had similar consequences, VZV infected MeWo cells were treated with pimecrolimus. As for the plaques of hyperfusogenic mutants gB[Y881F] and gH[TL] [18,25], the VZV pOka plaques had a narrower margin with pimecrolimus treatment at 4 days post infection (dpi) compared to medium and DMSO treated cells as visualized by immunohistochemistry (IHC) (**Fig 7A**). Plaque sizes of pOka were also significantly reduced by pimecrolimus ($0.68 \pm 0.32$ mm$^2$) compared to those of the medium ($1.34 \pm 0.49$ mm$^2$) or DMSO ($1.21 \pm 0.60$ mm$^2$) controls (**Fig 7B**). In addition, pimecrolimus-treated monolayers had more frequent microplaques, hypothesized to be due to premature detachment of cells from the centers of primary plaques also noted in MeWo cells infected with the gB[Y881F], and the transfer of VZV particles attached to cell membrane fragments to create secondary foci (**Fig 7D**). While these secondary plaques led to higher plaque frequency in pimecrolimus treated monolayers and an apparent equivalence of VZV growth kinetics at 2–4 dpi (**Figs 7D and S3**), the total area of plaques in pimecrolimus treated monolayers remained 50% less than that observed with medium or DMSO treatment (**Fig 7D**). Taken together, regulation of cell fusion by calcineurin was necessary for the typical propagation of VZV in cell monolayers during viral replication and was not limited to an effect on gB/gH-gL mediated cell fusion in the absence of other viral proteins.

The role of calcineurin phosphatase activity in viral spread was confirmed by quantifying plaque sizes in FKBP1A knockdown MeWo-DSP1 cells infected with VZV (**Fig 7C**). As expected, pimecrolimus treatment reduced the mean plaque size to 50% of that with medium or DMSO controls in both MeWo cells and MeWo-DSP1 control cells. In contrast, plaque sizes were not significantly decreased in FKBP1A knockdown MeWo-DSP1 cells treated with pimecrolimus compared to the medium and DMSO controls, indicating that exaggerated fusion due to the inhibition of calcineurin phosphatase activity was the primary factor in poor VZV spread.

## The phosphoproteome of MeWo cells treated with pimecrolimus revealed potential novel host calcineurin substrates

Calcineurin is known to be highly selective for its de-phosphorylation targets through recognition of two short docking motifs on its substrates, PxIxIT and LxVP [49,50]. The consensus

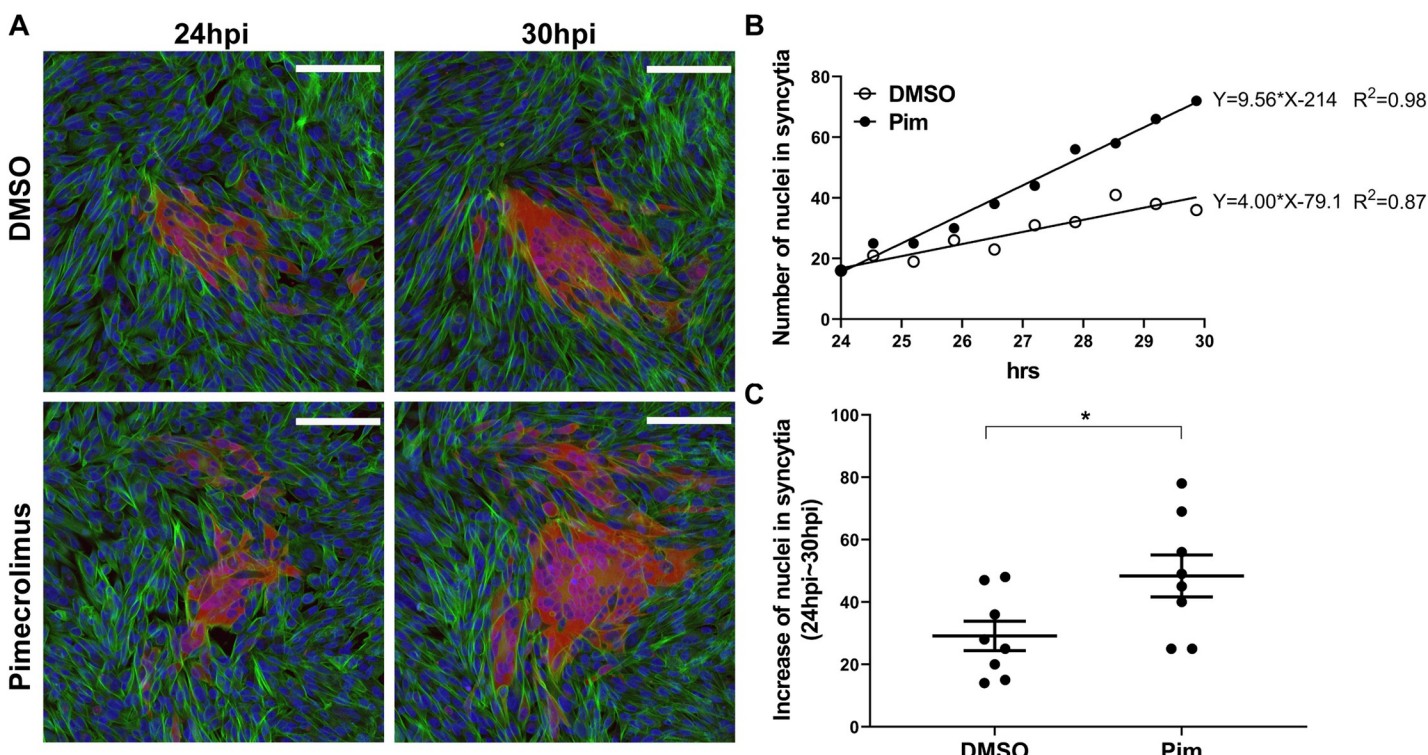

**Fig 6. Enhanced cell fusion induced by inhibition of calcineurin phosphatase activity during VZV infection.** (A) Confocal microscopy of LifeAct-tGFP MeWo cells infected with pOka-TK-RFP, treated with DMSO or pimecrolimus (Pim; 10 μM). Representative plaques captured from live-cell confocal microscopy at 24 and 30 hpi from **S1 and S2 Movies**, that show pOka-TK-RFP infected cells (red), LifeAct-tGFP labeled actin filaments (green), and nuclei stained with Hoechst 33342 (blue). Scale bars = 100 μm. (B) Frequency of nuclei in syncytia at 40 min intervals from 24 to 30 hpi for plaques in (A). Coefficient of determination ($R^2$) was calculated to determine the linear relationship of increased nuclei frequency in syncytia during the 6 hour window of infection. (C) Scatter dot plots of increased nuclei frequency in syncytia from 24 to 30 hpi. Mean ± SEM for plaques (n = 8) from two independent live-cell confocal microscopy experiment. Statistical significance was evaluated by an unpaired, nonparametric Mann-Whitney test (*, $p < 0.05$).

sequences for PxIxIT and LxVP motifs were derived from 20 previously identified calcineurin substrates (**Fig 8A and S3 Table**). The PxIxIT motif has a proline restriction at position 1 and only tolerates conservative replacements with hydrophobic amino acids at position 3 and 5, while position 6 is more degenerated, accepting both hydrophilic and hydrophobic residues. Similarly, the LxVP motif has a proline restriction at position 4 but tolerates conservative replacements with hydrophobic amino acids at position 1 and 3 (**Fig 8A**).

Therefore, as a first step towards defining potential downstream elements involved in fusion regulation, host substrates were investigated using phosphopeptide enrichment and Orbitrap mass spectrometry to identify the proteins that were exclusively phosphorylated when MeWo cells were treated with pimecrolimus compared to DMSO treated cells. Among 5,177 phosphoproteins detected, only seven were exclusively phosphorylated at serine or threonine sites in pimecrolimus-treated but not in DMSO-treated MeWo cells. Two of the seven were NFATC1 and ELK1, which are known targets of calcineurin dephosphorylation [50,51] (**Fig 8B and S2 Table**). Thus, the persistence of phosphorylated NFATC1 and ELK1 served as controls for the phosphoproteome data from pimecrolimus treated MeWo cells. Importantly, and to our knowledge, none of the remaining five proteins, PHACTR2, ILF3, NCOA1, TMC8, and MBD1, that remained phosphorylated in the presence of pimecrolimus have been reported previously as substrates for calcineurin. The four phosphoproteins, PHACTR2, ILF3, NCOA1 and TMC8 all contained the PxIxIT and LxVP consensus docking motifs, with

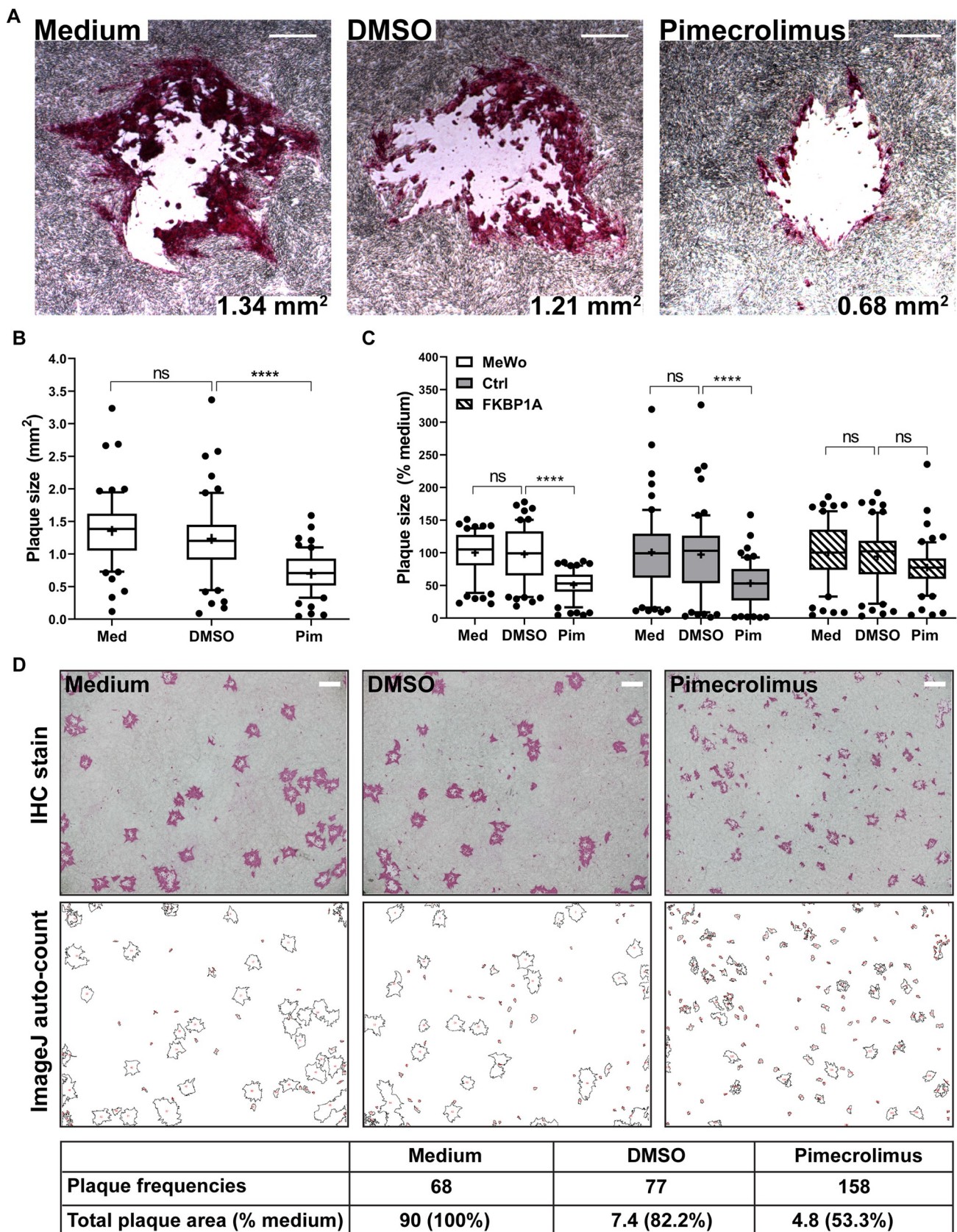

**Fig 7. Inhibition of calcineurin phosphatase activity suppresses VZV spread in MeWo cells.** Plaque sizes of VZV pOka in MeWo cells untreated (medium; Med), treated with DMSO or pimecrolimus (Pim; 10 μM) at 4 dpi. (A) Immunohistochemistry staining of VZV plaques with the mean plaque size per condition. Scale bars = 0.3 mm. (B) Box and whisker plots of plaque sizes (mm$^2$). Boxes show the 25–75 percentile, whiskers extend to 10–90 percentile, the median is the horizontal band, and the mean (+) from three independent experiments (n = 60). Dots are outliers. Statistical differences were evaluated by one-way ANOVA (ns, not significant; ****, $p < 0.0001$). (C) Box and whisker plots of MeWo cells, MeWo-DSP1 control cells or FKBP1A KD MeWo-DSP1 cells at 4 dpi with VZV pOka, untreated (medium; Med) or treated with DMSO or pimecrolimus (Pim, 10 μM). Percentage of plaque size normalized to the untreated (% medium). Boxes show the 25–75 percentile, whiskers extend to 10–90 percentile, the median is the horizontal band, and the mean (+) from two independent experiments (n = 60). Dots are outliers. Statistical differences were evaluated by two-way ANOVA (ns, not significant; ****, $p < 0.0001$). (D) Frequency and total area of plaques in the presence of pimecrolimus. MeWo cells infected with pOka were untreated (medium), treated with DMSO or pimecrolimus (10 μM) for 4 days, plaque frequency and total plaque area were analyzed by ImageJ automatic particle counting. Representative result from ≥3 independent experiments is shown, with input Immunohistochemistry images (scale bars = 2 mm), processed images and measurements listed.

conservative substitutions at positions of 3 and 5 on PxIxIT, and positions 1 and 3 on LxVP, suggesting they are likely direct calcineurin substrates (**Fig 8C**). However, MBD1 lacked a conservative substitution at position 3 on PxIxIT motif, which might weaken calcineurin docking (**Fig 8C**). If so, MBD1 phosphorylation might be indirectly due to calcineurin inhibition. Taken together, the inhibition of calcineurin phosphatase activity leads to unique phosphorylation of these seven host proteins and exaggerated VZV induced cell fusion. What role the dephosphorylation of one or more of these proteins plays in the calcineurin regulation of VZV cell fusion is unknown but the present study provides the basis for further investigations.

## Discussion

A hallmark of VZV pathogenesis is its ability to overcome the intrinsic barriers to cell fusion triggered by the gB/gH-gL complex. The present study utilized the HT-SRFA to identify a critical cell host factor, calcineurin, required for canonical VZV mediated cell fusion. We demonstrate that both tacrolimus and pimecrolimus, well-known calcineurin inhibitors, act by forming a drug-protein complex with FKBP1A which prevents calcineurin from functioning as a cellular phosphatase. In contrast to tacrolimus and pimecrolimus, sirolimus, a drug that also forms a complex with FKBP1A but inhibits mTOR instead of calcineurin, failed to prevent calcineurin activation induced by ionomycin and did not affect cell fusion induced by either VZV gB/gH-gL, HSV-1 gB/gH-gL/gD, or syncytin-1. This finding differentiates the two downstream effects that result from binding of these drugs to FKBP1A and strongly supports that calcineurin phosphatase activity is critical for the regulation of cell fusion induced by VZV gB/gH-gL and these other fusogens.

As a highly cell-associated virus, VZV releases very few infectious virus particles into the extracellular environment [52]. Thus, disrupted cell fusion affects VZV spread more than it does for other alphaherpesviruses, like HSV-1, which produces orders of magnitude more cell-free virions, even in the context of syncytial phenotypes [53–56]. Our previous studies have demonstrated that a base level of cell fusion is required for VZV spread as mutations of the gB or gH ectodomain that disrupt cell fusion can cause severe replication impairment or inactivate VZV, allowing little to no virus spread [18,21,22]. However, failure to control this virus-induced cell fusion process also markedly reduces virus spread as mutations in the gB or gH CTD that enhance fusion actually suppress virus spread due to impeded virus particle assembly [18,25]. Notably, the exaggerated cell fusion also impairs VZV infection in skin xenografts [18,25]. Therefore, in contrast to other alphaherpesviruses that produce large quantities of extracellular virions, VZV induced cell fusion needs to be tightly regulated to allow virus propagation for pathogenesis. The present study links calcineurin phosphatase activity to this regulation.

As one of the key proteins in calcium signaling networks, calcineurin is involved broadly in host cell gene expression, membrane trafficking, cytoskeleton arrangement, cell cycle, and

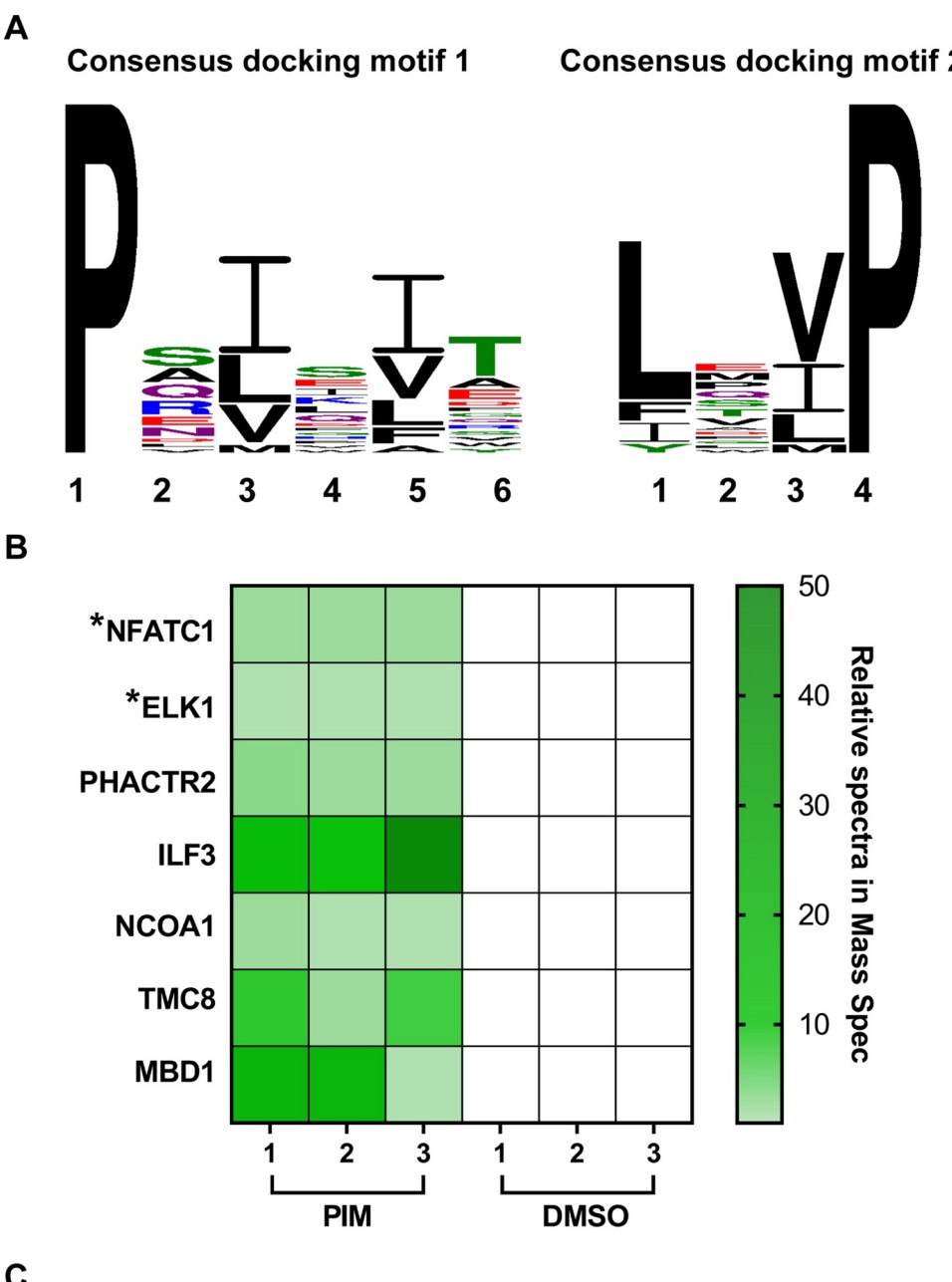

**Fig 8. Inhibition of calcineurin phosphatase activity alters the phosphoproteome of MeWo cells.** (A) Consensus docking motifs on calcineurin substrates for calcineurin binding. PxIxIT and LxVP motifs from 20 previously verified substrates of calcineurin were used to generate the consensus motif by WebLogo 3. Frequency of amino acids at each position is represented by the height of the letter. Chemical properties of amino acids are represented by the color of the letter: green, polar; purple, neutral; blue, basic; red, acidic; black, hydrophobic. (B) Heat map of uniquely phosphorylated proteins detected in MeWo cells treated with pimecrolimus (10 μM) for 2 hrs. Total cellular proteins were extracted in the presence of phosphatase inhibitor and subject to Zr-IMAC phosphopeptide enrichment, followed by orbitrap mass spectrometry. Three biological samples were analyzed for each condition and the relative spectra of uniquely phosphorylated proteins detected in all three pimecrolimus treated samples but not in DMSO samples are shown in the heat map. Nuclear factor of activated T cells cytoplasmic 1 (NFATC1); ETS transcription factor (ELK1); phosphatase and actin regulator 2 (PHACTR2); interleukin enhancer binding factor 3 (ILF3); nuclear receptor coactivator 1 (NCOA1); transmembrane channel-like protein 8 (TMC8); methyl-CpG binding domain protein 1 (MBD1). Previously known calcineurin substrates are highlighted with asterisks (*). (C) Potential docking motif for calcineurin binding on newly identified phosphoproteins located by Clustal alignment using the consensus motif. Conserved amino acids on PxIxIT and LxVP motif (red), conservative substitutions (blue), docking motif on previously known substrates of calcineurin (boxed).

apoptosis [57]. In this context, calcineurin has been implicated specifically in the fusion of myoblasts required for skeletal muscle differentiation and growth shown by the consequences of inhibiting its function [58,59]. Moreover, calcineurin regulates cell fusion needed for bone formation in that its inhibition by tacrolimus reduces the formation of differentiated multinucleated osteoclasts [60,61]. While other cell factors contribute to controlling virally induced cell-cell fusion, for example, IFITM, tetraspanin, Ezrin, and EWI-2, which were shown to inhibit T cell syncytia induced by HIV [62–66], to our knowledge, calcineurin has not been reported to regulate cell fusion induced by any of the fusion inducing viruses. The finding that inhibition of calcineurin phosphatase activity also enhanced cell fusion induced by HSV-1 gB/gH-gL/gD highlights that calcineurin has a general role in viral fusion regulation. That VZV employs the phosphatase activity of this host cell protein in addition to the direct viral regulation by functional motifs in the gB and gH CTDs to prevent exaggerated cell fusion reflects a well evolved homeostasis needed for progeny virion assembly, which is disrupted when fusion is not regulated effectively.

In an initial exploration of potential calcineurin substrates that might be involved in its fusion regulation function, the present study identified seven host phosphoproteins that remained phosphorylated when calcineurin phosphatase activity was inhibited by pimecrolimus (**S2 Table**). The mRNA transcripts for these five proteins (PHACTR2, ILF3, NCOA1, TMC8, and MBD1) as well as the two known calcineurin substrates (NFATC1 and ELK1) were all detected in VZV infected cells at comparable levels to uninfected cells [33]. Of these, nuclear receptor co-activator 1 (NCOA1), a member of the steroid/nuclear receptor coactivator family, is of interest because its interactome includes estrogen receptor (ER), retinoic acid receptor (RAR) and retinoid X receptor RXR (RXR) [67–70]. The HT-SRFA identified three compounds that are estrogen receptor antagonists and two compounds that are retinoic acid receptor ligand or retinoic acid receptor agonists, all of which enhanced cell fusion (**S1 Table**). The interaction between NCOA1 and ER, RAR or RXR depends primarily on the LxxLL motif residing in the nuclear receptor interaction domain on NCOA1 [71]. Ser-369 is in a serine/threonine rich region of NCOA1, while Ser-698 is four amino acids downstream of the fourth LxxLL motif on NCOA1, which might affect the conformation of NCOA1 in a way that alters the accessibility of the nuclear receptor [72]. Phosphorylation at these sites due to calcineurin inhibition (**S2 Table**) would be expected to have an analogous effect on fusion as observed with drugs identified by the HT-SRFA that target the nuclear receptors directly. PHACTR2 is also of particular interest because members of the phosphatase and actin regulators (PHACTR1-4) family are known for their role in the modulation of cytoskeleton structure [73]. Monomeric G-actin binds to PHACTR4; when monomeric G-actin level becomes low,

PHACTR4 senses the change and promotes the depolymerization of actin filaments to replenish the monomeric G-actin pool [74]. PHACTR2 is less studied but might regulate actin dynamics in a similar fashion since all PHACTR proteins share sequence similarity in G-actin binding domains [73]. Ser-510 on PHACTR2 was first detected to be phosphorylated in response to calcineurin inhibition (**S2 Table**), and can be mapped to the region near where G-actin binds. Phosphorylation at this site might hinder the interaction between PHACTR2 and G-actin, thus triggering actin depolymerization. This could potentiate cell fusion given that the actin cortex has been reported to control syncytium formation by inhibiting fusion pore expansion [75], and is consistent with the enhanced VZV cell fusion induced in the absence of calcineurin phosphatase activity.

In addition to VZV, cell fusion is also a component of HSV pathogenesis, evident from the multinucleated cells observed in infected skin tissues [2]. Of interest, a recent study linked protein tyrosine phosphatase 1B (PTP1B) to HSV-1 induced cell-cell fusion through observations with a fusion enhancing compound salubrinal [76]. However, in contrast to the enhanced VZV gB/gH-gL dependent cell fusion due to calcineurin inhibition, salubrinal-induced fusion of HSV-1 infected cells was dependent on viral accessory proteins as PTP1B and the target of salubrinal did not regulate the core fusion machinery in isolation but only when it was in a complex with accessory proteins. In addition, the persistent phosphorylation of eIF2α, a well-established host response to salubrinal, still occurred when PTP1B was knocked out, indicating that PTP1B was not the direct target of salubrinal, and the effect of salubrinal on HSV-1 cell fusion likely involved additional, yet unidentified proteins. This study highlighted a multifaceted interplay between host cell proteins in the regulation of HSV-1 cell fusion.

Unsurprisingly, owing to the multi-factorial regulation of the complex biology of herpesvirus induced cell fusion, the HT-SRFA data identified additional host cell proteins that might also be involved in fusion regulation independently of the calcineurin pathway and warrant further study. Differential regulation that depends on cell-type specific expression of host cell factors might also explain the fact that while VZV triggers cell fusion between skin cells and in sensory ganglia, VZV-infected T cells do not fuse [77–79]. Since VZV induced cell fusion is pivotal for pathogenesis in the human host, host-directed therapies that target fusion could improve antiviral treatment, and by extension, such factors present opportunities for the design of interventions against other fusogenic viral pathogens.

## Materials and methods

### Cells and viruses

CHO-DSP1 [28] cells, derived from Chinese Hamster Ovary (CHO) K1 cells (CCL-61, ATCC) that express Dual Split Protein (DSP)$_{1-7}$ of the chimeric reporter protein composed of split GFP and split *Renilla* luciferase, were propagated using F-12K nutrient mixture medium (Corning) supplemented with 10% fetal bovine serum (FBS, Gibco), penicillin (100 U/ml, Gibco), and streptomycin (100 μg/ml, Gibco), with puromycin selection (8 μg/ml; Gibco). Human melanoma MeWo cells (HTB-65, ATCC), MeWo-DSP1 cells derived from MeWo cells that express DSP$_{1-7}$, [28], MeWo-DSP2 expressing the DSP$_{8-11}$ [28], MeWo-DSP1 cells expressing shRNA against FKBP1A, MeWo-DSP1 cells expressing shRNA against CNB1, and MeWo LifeAct-tGFP cells expressing a short peptide fused with tGFP that binds to F-actin [33] were propagated in minimal essential medium (MEM, Corning) supplemented with 10% FBS, non-essential amino acids (1X, Corning), penicillin (100 U/ml, Gibco), streptomycin (100 μg/ml, Gibco), and amphotericin B (0.5 μg/ml, Corning). MeWo-DSP1, MeWo-DSP2, MeWo-DSP1 expressing shRNA against FKBP1A, MeWo-DSP1 expressing shRNA against CNB1, and MeWo LifeAct-tGFP cells were kept under puromycin selection (5 μg/ml; Gibco). HEK293T

(CRL-3216, ATCC) cells were propagated using Dulbecco's modified Eagle's medium (DMEM, Corning) supplemented with 10% FBS, penicillin (100 U/ml, Gibco), streptomycin (100 μg/ml, Gibco) and amphotericin B (0.5 μg/ml, Corning). The parental Oka strain of VZV (pOka) derived from a self-excisable bacterial artificial chromosome (BAC) [80] and the pOka-TK-RFP virus engineered to express monomeric RFP conjugated to the thymidine kinase (TK) [33] were propagated, and titrated on MeWo cells.

### High-throughput stable reporter fusion assay

NIHCC (NIH clinic collection 446 compounds, Evotec), LOPAC (Library of Pharmacologically Active Compounds 1280 compounds, Sigma Aldrich), Microsource spectrum (2000 compounds, Discovery System, Inc.), SCREEN-WELL ICCB Known Bioactives Library (480 compounds, Enzo Life Sciences), and SCREEN-WELL FDA approved drug library (640 compounds, Enzo Life Sciences) available from the High-Throughput Bioscience Center (HTBC) at Stanford University, total of 4846 compounds, were evaluated for their effects on VZV gB/gH-gL mediated cell fusion. Briefly, $7.2 \times 10^7$ CHO-DSP1 cells were transfected with 144 μg each of pCAGGS-gB [27], pME18S-gH[TL] [27], and pcDNA3.1-gL [22] plasmids using Lipofectamine 2000 (Invitrogen). The gH[TL] vector lacking the last eight amino acids (834–841) in the cytoplasmic domain was selected over wild type gH because gH[TL] provides an enhanced signal-to-noise ratio. CHO-DSP1 cells were chosen over MeWo-DSP1 cells as the effector cells because CHO cells attain higher transfection rate, which increases the percentage of effector cells that express all three glycoproteins together, thus increases the chance of cell fusion event. While transfection was underway, libraries of compounds were pinned into wells of 384-well plates (Greiner Bio-One) preloaded with 30 μl/well medium in the HTBC facility. At 6 hrs post-transfection, the transfected CHO-DSP1 cells were harvested and resuspended in 45 ml MEM medium to achieve $1.6 \times 10^6$ cell/ml, and mixed with four volumes (180 ml) of MeWo-DSP2 cells at $1.5 \times 10^6$ cell/ml. The mixed cell culture was added into 384-well plates at 30 μl/well using the automated dispenser system (WellMate Microplate Reagent Dispenser with Stackers, Matrix). Cells were incubated at 37˚C for 48 hrs before the measurement of *Renilla* luciferase activity from the reconstitution of DSP due to cell fusion. Given the large number of 384-well plates in this experiment, the total time for plate reading would be up to 50 min, which exceeds the stable luminescence time window (2 min) generated by traditional coelenterazine substrate. Enduren live cell substrate (Promega) was chosen as an alternative substrate because it can generate sustained luminescence signal up to 24 hrs. For readout, 20 μl/well 1X Enduren live cell substrate was dispensed into wells using automated dispenser system (Multidrop 384-well Microplate Reagent Dispenser, Titertek), followed by 2 hrs incubation at 37˚C. *Renilla* luminescence was then detected using Infinite M1000 PRO-Multimode 384-well Microplate Reader (Tecan). Cell viability was also tested dispensing 10 μl/well 1X CellTiter-Glo luminescent substrate (Promega), and the resultant luminescence was measured as above. Positive controls of the assay were wells containing cells but without any compound treatment (n = 512). Transfection with vehicle plasmids pcDNA3.1 (+) (Invitrogen) and pME18S [27] served as a negative control for fusion (n = 256), while wells with medium only as a negative control for cell viability (n = 256). The experiment was performed in duplicate.

HT-SRFA data was analyzed using Microsoft Excel. The average Z' values of *Renilla* luminescence and CellTiter-Glo luminescence from duplicate experiments were 0.53 ± 0.08, and 0.67 ± 0.10 respectively, validating a good separation between the positive and negative controls in both readings, and the quality of the assay was in the excellent range ($1 > Z' \geq 0.5$) for high-throughput screening assays [81]. The values of *Renilla* luminescence and CellTiter-Glo luminescence values for test compound wells were normalized to and presented as a

percentage of the mean of the intraplate positive control wells. The average coefficient of variation (CV %) of positive controls on fusion and cell viability was 14.9% and 9.9% respectively. Compounds with a cell viability value outside the ±3 CV % ($\geq$129.7% and $\leq$ 70.3%) were eliminated as erroneous cell dispensing or potentially toxic, thus the screen initially identified 167 compounds that significantly affected gB/gH-gL dependent cell fusion based on a reproducible increase or decrease of more than ±3 CV % of fusion ($\geq$ 144.7% or $\leq$ 55.3%); 95 compounds enhanced and 72 inhibited fusion. However, since CHO-DSP1 cells comprised 20% of the cells/well, a compound toxic only for CHO cells might be mis-identified as inhibitory to fusion because 20% cell death was below the cell viability cutoff. Therefore, the CHO cell toxicity profiles for the 72 compounds classified as inhibitors were determined from the HTBC database, which led to the elimination of 58 compounds. When compounds that were included more than once in the libraries were further removed, 71 (1.5%) unique compounds were found to enhance fusion and 9 (0.2%) inhibited fusion.

## Validation of compounds from HT-SRFA

Tacrolimus (FK506) (Selleckchem), pimecrolimus (Selleckchem), sirolimus (Selleckchem), and ionomycin (Alomone labs) were all dissolved in DMSO (Sigma Aldrich). Activity of the compounds on VZV gB/gH-gL mediated cell fusion were evaluated by stable reporter fusion assay as described previously [28]. Briefly, CHO-DSP1 or MeWo-DSP1 cells transfected with equal quantities of pCAGGS-gB, pME18S-gH[TL], and pcDNA3.1-gL plasmids using Lipofectamine 2000 were harvested at 6 hrs post-transfection, and mixed with MeWo-DSP2 cells in the presence of various concentrations of the compounds prepared in two-fold serial dilutions, ranging from 10 μM to 1.25 μM. Co-culture of cells were seeded into Nunc MicroWell 96-well Optical-Bottom Plates (ThermoFisher) and incubated for 48 hrs. The activity of *Renilla* luciferase was read immediately after adding substrate h-Coelenterazine (Nanolight Technology). Transfection with only vehicle plasmids pcDNA3.1 (+) and pME18S, or pcDNA3.1 (+), pME18S and pCAGGS-gB served as negative control. Cell viability was measured using CellTiter-Glo Luminescent substrate (Promega). Luminescence signal was recorded using Synergy H1 Hybrid Multi-Mode Reader (BioTek). Experiments were performed at least in triplicate.

## Stable-reporter fusion assays

Assay dependent effector cells, CHO-DSP1, MeWo-DSP1, FKBP1A knockdown MeWo-DSP1, or CNB1 knockdown MeWo-DSP1, were transfected with VZV pCAGGS-gB, pME18S-gH[TL], and pcDNA3.1-gL plasmids, or HSV-1 pCAGGS-gB, pME18S-gH, pcDNA3.1-gL, and pCAGGS-gD plasmids, or pHCMV-ERVW1Δ16 (gift from George Murphy, Stanford University, Stanford, CA, USA) plasmid with Lipofectamine 2000. ERVW1Δ16 plasmid, which encodes a form of human syncytin-1 with a sixteen amino acids truncation from the cytoplasmic domain, was selected because it can generate a better signal-to-noise ratio in the fusion readout [82]. At 6 hrs post-transfection, the transfected cells were harvested and mixed with MeWo-DSP2 cells in the presence of DMSO, pimecrolimus (10 μM), or sirolimus (10 μM) for additional 48 hrs. The activity of *Renilla* luciferase was read immediately after adding substrate h-Coelenterazine (Nanolight Technology) on a Synergy H1 Hybrid Multi-Mode Reader (BioTek).

## Quantitation of cell surface gB and gH-gL

The quantity of cell surface VZV glycoproteins were measured as described previously [18]. Briefly, CHO-DSP1 cells transfected with either pCAGGS-gB, or pME18S-gH[TL] and pcDNA3.1-gL plasmids were treated with medium, DMSO, or pimecrolimus (10 μM) for 24

hrs. Cells were dislodged using an enzyme-free cell dissociation buffer (Gibco), fixed with 1% paraformaldehyde (PFA, Boston Bioproducts), and resuspended in FACS staining buffer [PBS with 0.2% IgG-free bovine serum albumin (Jackson ImmunoResearch), and 0.1% NaN$_3$ (Sigma Aldrich)]. Cell surface gB or gH (representing gH-gL heterodimer) were detected with primary antibodies of either mouse anti-VZV gB mAb (SG2-2E6, GeneTex #GTX38718), or mouse anti-VZV gH mAb (SG3, GeneTex #GTX40374), each at 1:100 dilution, followed by anti-mouse IgG-Alexa Fluor 555 (ThermoFisher). Total amounts of viral glycoprotein production was performed by the same staining protocol, except cells were permeabilized using Cytofix/Cytoperm kit (BD Biosciences) before adding the primary antibodies and during the staining procedure. Stained cells were analyzed using a DXP multi-color FACScan analyzer (Cytek Biosciences), and data were processed with FlowJo (TreeStar) to determine the quantity of total and surface levels of gB and gH respectively. The ratio of cell surface to total quantity was also calculated. Experiments were performed in triplicate.

## Construction of FKBP1A and CNB1 shRNA knockdown cell line

The miR30-based shRNA expression with pGIPZ lentiviral vector packaging system was used. Two pairs of shRNAs cloning primers targeting the FKBP1A (encoded by FKBP1A gene) or CNB1 (encoded by PPP3R1 gene) transcripts were designed using sigma bioinformatics website (http://www.sigmabioinfo.com/Informatics_tools/batch-search.php#shRNA). The first PCR step using AccuPrime Pfx DNA polymerase (Invitrogen) generated shRNA oligos. The second PCR step added on the remaining sequence of the miR30 cassette and also cloning restriction enzyme sites using Adapter-miR30PCRXhoI_Forward [28] and Linker-miR30linkerNotI_Reverse, or Linker-miR30linkerNotI_Forward and Adapter-miR30PCREcoRI_Reverse [28]. The final PCR products were resolved on a 4% agarose gel, purified by QIAquick gel extraction kit (Qiagen), and digested with XhoI/NotI or NotI/EcoRI (New England Biolabs). The pGIPZ-DSP1 vector reported previously [28] was also digested with XhoI/EcoRI, dephosphorylated with Antarctic phosphatase (New England Biolabs), separated on a 0.8% agarose gel, and purified as above. The purified products were ligated together using T4 DNA ligase (New England Biolabs), and transformed into TOP10F' Electrocomp E.coli cells (Invitrogen). Cells were then plated on LB plates supplemented with antibiotics ampicillin (100 µg/ml, Sigma Aldrich) and Zeocin (25 µg/ml, Gibco). Correct clones were confirmed by restriction enzyme digestion using MluI/XhoI (New England Biolabs), followed by sequencing using pGIPZ-shRNA-sequencing [28]. All the primers used above were listed in **S4 Table.**

Lentiviruses were generated by transfecting 1.2 10$^6$ HEK293T cells with 2.5 µg pGIPZ-DSP1 vector or pGIPZ-DSP1 vector containing the FKBP1A or CNB1 shRNAs constructed above, 2.5 µg psPAX2 (Addgene #12260), and 1 µg pMD2.G (Addgene #12259) plasmids using Lipofectamine 2000. At 24 hrs post-transfection, virus-containing supernatant was harvested and filtered through a 0.45 µm syringe filter (Fisher Scientific). 1.5 10$^6$ MeWo cells were then transduced with 1ml supernatant of lentivirus along with polybrene (4 µg/ml, Sigma Aldrich) by centrifuging the cells at 750 Relative Centrifugal Force (RCF) for 45min at room temperature to generate MeWo-DPS1 control cells, FKBP1A knockdown MeWo-DSP1 or CNB1 knockdown MeWo-DSP1 cells. Puromycin (5 µg/ml; Gibco) was added at 24 hrs post-transduction to select for the successfully transduced cell population, and cells were propagated in the presence of puromycin hereon.

## RT-qPCR

RNA from the cells was harvested using RLT buffer (Qiagen) and isolated using the QIAshredder columns (Qiagen) and the RNeasy Plus mini kit (Qiagen). cDNA was generated from the

isolated RNA using the SuperScripIII first-strand synthesis system (Invitrogen). Quantitative PCR (qPCR) was performed with the cDNA using SsoAdvanced Universal SYBR green super-mix (BioRad) and the qPCR primers (S4 Table), on CFX384 real-time PCR detection system (BioRad) in triplicate. The results were normalized to house-keeping gene, PGM1, and comparisons were analyzed based on ΔΔCt using the CFX Manager (BioRad).

## His-tagged FKBP1A pulldown

CHO-DSP1 cells were transfected with control plasmid pcDNA 3.1 (+) or plasmid pcDNA3.1_huFKBP1A-His-Myc [83]. At 24 hrs post-transfection, cells were lysed in lysis buffer [50 mM $NaH_2PO_4$ (Fisher Scientific), 300 mM NaCl (Fisher Scientific), 10 mM imidazole (Sigma Aldrich), 0.05% Tween20 (Sigma Aldrich), pH 8.0] with cOmplete EDTA-free protease inhibitor (Sigma Aldrich) for 30 min on ice. The sample was centrifuged at 3,000 RCF for 10 min at 4°C to remove cell debris, and the supernatant was saved as cell extract. Ten volumes of cell extract were precipitated with 1 volume of pre-equilibrated Ni-NTA agarose beads (Qiagen) at 4°C overnight, followed by washes using wash buffer [50 mM $NaH_2PO_4$, 300 mM NaCl, 20 mM imidazole, 0.05% Tween20, pH 8.0]. Meanwhile, MeWo-DSP1 cells were extracted using the above lysis buffer and treated with DMSO, 10 μM of tacrolimus, 10 μM pimecrolimus or 10 μM sirolimus for 30 min at 4°C with end-over-end rotation, followed by incubation with the above-prepared FKBP1A-retaining Ni-NTA agarose beads at 4°C overnight. The bound proteins were eluted from the beads using elution buffer [50 mM $NaH_2PO_4$, 300 mM NaCl, 250 mM imidazole, 0.05% Tween20, pH 8.0], followed by western blotting analysis.

## Western blotting

Cell pellets collected by centrifugation at 500 RCF for 5 min were lysed in extraction buffer [0.1 M NaCl, 5 mM KCl (Fisher Scientific), 1 mM $CaCl_2$ (Fisher Scientific), 0.5 mM $MgCl_2$ (Fisher Scientific), 1% IGEPAL CA-630 (Sigma Aldrich), 1% Deoxycholate (Sigma Aldrich) in 0.1 M Tris buffer, pH 7.2], with cOmplete EDTA-free protease inhibitor, for 30 min on ice. The sample was centrifuged at 3,000 RCF for 10 min at 4°C to get rid of cell debris, and the supernatant was saved as cell extract. Cell extract was 1:1 diluted in 2X Laemmli sample buffer (BioRad) supplemented with 5% reducing reagent β-mercaptoethanol (Sigma Aldrich), boiled at 95°C for 5 min, and separated on a precast 4~20% gradient SDS-PAGE gel (BioRad). Proteins were then transferred to Immobilon-P PVDF membranes (Millipore) using Trans-blot SD Semi-dry Transfer cell (BioRad) in transfer buffer [48 mM Tris Base (Fisher Scientific), 39 mM Glycine (Fisher Scientific) and 0.01% SDS (Sigma Aldrich), with 20% methanol (Sigma Aldrich)]. The transferred proteins were probed with primary rabbit polyclonal anti-human FKBP1A (ThermoFisher #PA1-026A, 1:2,500), mouse monoclonal anti-calcineurin B1 (Sigma Aldrich #C0581, 1:3,000), mouse monoclonal anti-human α-Tubulin (Sigma Aldrich #T5168, 1:5,000), or rabbit polyclonal anti-human calcineurin A (Cell Signaling #2614S, 1:1,000) diluted in probing buffer [20 mM Tris Base, 150 mM NaCl, 0.1% Tween20 with 1% IgG-free BSA, pH 7.6], followed by incubation with anti-rabbit or anti-mouse IgG horseradish peroxidase-linked secondary antibodies (GE Healthcare Life Sciences, UK Ltd., 1:10,000) and Pierce ECL Plus Western Blotting Substrate (ThermoFisher). Chemiluminescence was detected using BioMax MR Film (Carestream). Band density analysis was performed using ImageJ.

## Live-cell confocal microscopy

LifeAct-tGFP MeWo cells were seeded into NuncLab-Tek II Chambered Coverglass (Thermo-Fisher) at 2 x$10^5$ cell/cm$^2$ 24 hrs prior to infection with pOka-TK-RFP. Treatment with DMSO

or pimecrolimus (10 μM) were started at 2 hpi. At 24 hpi, the medium was changed to contain 2 μg/ml Hoechst 33342 (ThermoFisher) in addition to DMSO or pimecrolimus (10 μM). Live imaging was performed immediately at the Stanford Cell Sciences and Imaging Facility (CSIF) using a LSM 880 confocal microscope (Zeiss). At 20X magnification, foci of infection were captured every 8 min for 16 hrs. Time-lapse movies were created using FFmpeg (FFmpeg Developers, (2016), ffmpeg tool (Version be1d324) [Software]. Available from http://ffmpeg.org/).

## NFATC1 nuclear translocation assay by fluorescence microscopy

MeWo cells, MeWO-DSP1 cells, or FKBP1A knockdown MeWo-DSP1 cells ($4 \times 10^5$) seeded onto coverglass (Fisher Scientific 18CIR-1) were transfected with 1 μg of EGFPC1-huNFAT-c1EE-WT plasmid (Addgene #24219, deposited by Jerry Crabtree) [84]. At 24 hrs post-transfection, cells were treated with medium, DMSO, ionomycin (1 μM), tacrolimus (10 μM), pimecrolumus (10μM) or sirolimus (10 μM) for 30 min, or pretreated with tacrolimus (10 μM), pimecrolumus (10μM) or sirolimus (10 μM) for 30 min prior to additional 30 min treatment of ionomycin (1 μM). For testing NFATC1 translocation in VZV infected cells, MeWo cells ($2 \times 10^6$) seeded into 6-well plates were transfected with 5 μg of the EGFPC1-huN-FATc1EE-WT plasmid and trypsinized at 6 hrs post-transfection to seed on the coverglass, then inoculated with pOka-TK-RFP virus for additional 16 hrs before drug treatment. Cells were then fixed with 4% PFA and stained with 10 μg/ml Hoechst 33342. Images were captured on BZ-X710 fluorescence microscope (Keyence) at 100X oil lens. Channel merging were performed using ImageJ.

## VZV plaque size assay

Monolayers of MeWo cells ($1 \times 10^5$ cell/cm$^2$) were inoculated with 5 plaque forming unit (PFU) per cm$^2$ of cell-associated pOka viruses for 2 hrs, followed by addition of medium, DMSO, or pimecrolimus (10 μM). The drug-containing medium was changed every 48 hrs before the cells were fixed with 4% PFA at 4 days post-infection. Immunohistochemistry (IHC) was performed on the fixed cells using mouse monoclonal anti-VZV antibody (Mixed, GeneTex #GTX38720, 1:2000 diluted in PBS), followed by incubation with biotinylated anti-mouse IgG (Vector Laboratories #BA-9200), and alkaline phosphatase streptavidin (Jackson ImmunoResearch #016-050-084). The plaques were visualized by staining with a mixture of FastRed salt (Sigma Aldrich) and Naphthol AS-MX phosphate (Sigma Aldrich). Images of plaques were taken with Axio microscope (Zeiss) using 2.5X lens. The plaques were outlined and the area of plaque was calculated using Image J. Monolayers of infected MeWo cells processed with IHC stain was also taken by BZ-X710 fluorescence microscope (Keyence) at 2X lens, and plaque frequency and total infected area were analyzed by ImageJ automatic particle counting. Experiments were performed at least three times.

## VZV growth curve analysis

Monolayers of MeWo cells ($1 \times 10^5$ cell/cm$^2$) were inoculated with 100 PFU per cm$^2$ of cell-associated pOka viruses for 2 hrs, followed by addition of medium, DMSO, or pimecrolimus (10 μM). The drug-containing medium was changed every 48 hrs. Infected cells were harvested at every 24 hrs intervals till 4 days post-infection, and were titrated on fresh monolayers of MeWo cells in triplicate, followed by fixation with 4% PFA at 4 days post-infection and then IHC staining.

## Phosphopeptide enrichment mass spectrometry

MeWo cells were treated with vehicle control DMSO or pimecrolimus (10 μM) for 2 hrs before total cellular proteins were extracted by 2% SDS, Tris buffer, pH 7.6, in the presence of phosphatase inhibitors [20 mM NaF (Sigma Aldrich), 1 mM $Na_3VO_4$ (Sigma Aldrich), and 1mM β-glycerol (Sigma Aldrich)]. Three biological replicate samples for each condition were analyzed by Zr-IMAC phosphopeptide enrichment, followed by Orbitrap mass spectrometry performed in Stanford University Mass Spectrometry facility.

## Phosphoproteome analysis

Peptide and protein identification was conducted using Byonic V3.5.0 (Protein Metrics) at 1% false discovery rate. Uniquely phosphorylated proteins in pimecrolimus-treated samples but not in DMSO-treated samples were identified using Uniphosphoprotein-finder code generated for this paper using MatLab (Mathworks). The consensus calcineurin docking motifs from known calcineurin substrates were generated by WebLogo 3 [85,86], while potential docking motifs on newly identified phosphoproteins were located by Clustal Omega (EMBL-EBI) alignment using consensus docking motifs.

## Supporting information

**S1 Fig. The effect of pimecrolimus on cell fusion depends on gB and gH/gL complex.** Co-culture of target cells MeWo-DSP2 with effector cells CHO-DSP1 (A) or MeWo-DSP1 (B) transfected with plasmids expressing VZV gB/gH[TL]-gL (gB/gH-gL), empty vectors (Vector) or plasmid expressing gB only (gB) were untreated (medium) or treated with DMSO, or tacrolimus (10 μM), pimecrolimus (10 μM), sirolimus (10 μM) for 48 hrs. Cell fusion efficiency was measured and normalized to that of effector cells transfected with gB/gH[TL]-gL and untreated (% gB/gH-gL medium). Mean ± SEM represent ≥ 3 independent experiments. Statistical differences were assessed by comparison of values to that of effector cells transfected with gB and treated with DMSO using one-way ANOVA (ns, not significant; ****, $p < 0.0001$).
(TIF)

**S2 Fig. Effects of pimecrolimus on the expression and trafficking of gB and gH/gL.** (A) CHO-DSP1 cells transfected with plasmids expressing VZV gB or gH[TL]/gL, were untreated (medium) or treated with DMSO, or pimecrolimus (10 μM) for 24 hrs. The total, and cell surface expression of gB or gH was analyzed by flow cytometry of immunostained permeabilized or nonpermeabilized cells. The population of cells that express gB or gH as total or on the cell surface were normalized to the untreated (% medium) respectively. Mean ± SEM from three independent experiments are shown. (B) The expression level of gB or gH from (A) at cell surface were normalized to the total expression level and presented as a percentage of the total. The brackets represent the statistical differences evaluated by two-way ANOVA (ns, not significant; ***, $p < 0.001$; ****, $p < 0.0001$).
(TIF)

**S3 Fig. VZV replication kinetics in the presence of pimecrolimus.** MeWo cells infected with pOka were untreated (medium), treated with DMSO or pimecrolimus (10 μM) for 4 days; media was changed every 48 hrs. Monolayers of infected cells were harvested and titrated on fresh MeWo cells to determine PFU/ml. Representative result from two independent experiments is shown, with mean ± SEM analyzed by two-way ANOVA (**, $p < 0.01$).
(TIF)

**S1 Movie. Record of VZV syncytia formation in the presence of DMSO by time-lapse live-cell confocal microscopy.** LifeAct-tGFP MeWo cells infected with pOka-TK-RFP treated with vehicle control DMSO were subject to time-lapse live-cell confocal microscopy between 24~40 hpi, with capture taken place every 8 min. Infected cells detected by RFP (red), actin filaments by LifeAct-tGFP (green), and nuclei stained with Hoechst 33342 (blue). Frame rate: 5 frame/second.
(MP4)

**S2 Movie. Record of VZV syncytia formation in the presence of pimecrolimus by time-lapse live-cell confocal microscopy.** LifeAct-tGFP MeWo cells infected with pOka-TK-RFP treated with pimecrolimus (10 μM) were subject to time-lapse live-cell confocal microscopy between 24~40 hpi, with capture taken place every 8 min. Infected cells detected by RFP (red), actin filaments by LifeAct-tGFP (green), and nuclei stained with Hoechst 33342 (blue). Frame rate: 5 frame/second.
(MP4)

**S1 Table. Compounds from HT-SRFA with significant effect on VZV gB/gH-gL mediated fusion.** [C] indicates the concentration of compounds used in the HT-SRFA, fusion (%) is the measurement of *Renilla* luciferase activity normalized to that of the intraplate positive controls (no drug). Average from duplicate experiment is shown.
(XLSX)

**S2 Table. Uniquely phosphorylated proteins detected in pimecrolimus treated MeWo cells by Zr-IMAC phosphopeptide enrichment mass spectrometry.** Phosphoproteins exclusively present in three independent samples from pimecrolimus treated MeWo cells (Pim1, Pim 2 and Pim 3) were shown. Cleavage sites are indicated by the black dot. Serine (S[+80]) or threonine (T[+80]) phosphorylation modifications are highlighted in red. M[+16] indicates methionine oxidation, and C[+71] indicates cysteine propionamide modification.
(XLSX)

**S3 Table. Calcineurin docking motifs residing on known substrates of calcineurin.**
(DOCX)

**S4 Table. Primers for shRNA expressing cell construction and RT-qPCR analysis.**
(DOCX)

## Acknowledgments

We thank the Stanford High-throughput Bioscience Center for assistance with the HT-SRFA and Edward Yang (previous member of Arvin lab) for early phase development; the Stanford Cell Sciences Imaging Facility for assistance with live-cell confocal microscopy; the Vincent Coates Foundation Mass Spectrometry Laboratory, Stanford University Mass Spectrometry for conducting Zr-IMAC phosphopeptide enrichment mass spectrometry.

## Author Contributions

**Conceptualization:** Momei Zhou, Vivek Kamarshi, Ann M. Arvin, Stefan L. Oliver.

**Funding acquisition:** Ann M. Arvin.

**Investigation:** Momei Zhou, Vivek Kamarshi.

**Methodology:** Momei Zhou, Vivek Kamarshi, Ann M. Arvin, Stefan L. Oliver.

**Supervision:** Ann M. Arvin, Stefan L. Oliver.

**Writing – original draft:** Momei Zhou.

**Writing – review & editing:** Momei Zhou, Ann M. Arvin, Stefan L. Oliver.

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
