## [Decision Letter · Decision Letter 0]

27 Jul 2020

Dear Dr Zhou,

Thank you very much for submitting your manuscript "Calcineurin phosphatase activity regulates Varicella-Zoster Virus induced cell-cell fusion" for consideration at PLOS Pathogens. As with all papers reviewed by the journal, your manuscript was reviewed by members of the editorial board and by several independent reviewers. In light of the reviews (below this email), we would like to invite the resubmission of a significantly-revised version that takes into account the reviewers' comments.

As you will see, there was a range of opinions by the reviewers, with reviewer 3 being highly positive and reviewer 1 suggesting a series of necessary additional experiments and rewriting that they considered essential. There was a range of recommendations by the reviewers, who all recognized the potential interest to the field, but raised some substantial concerns about the manuscript as it currently stands. These issues must be addressed before we would be willing to consider a revised version of your study. We cannot, of course, promise publication at that time. We therefore ask you to modify the manuscript according to the review recommendations before we can consider your manuscript for acceptance. Your revisions should address the specific points made by each reviewer.

We feel that it is particularly important for you to address the activities of calcineurin and its downstream targets in VZV infected cells. You will notice that while all three reviewers were highly positive in the assessment and clarity of the manuscript (although they did have suggestions to increase the clarity of the methods used and provide a more detailed explanation of the approaches that I think you will want to consider), a theme that did run through the reviews was the consideration of the importance of calcineurin and its newly identified downstream targets to VZV infection. These were the subject of most of the rather speculative discussion. Reviewer 1 felt the study was not incomplete without at least some analyses of these candidates for their role in VZV infection, and reviewer 2 in comments 8 also outlined that the studies were not well validated in VZV infected cells and should be to some extent. Reviewer 3 communicated to the editor privately that this was an issue that they considered in their evaluation and recommendation. You may want to consider doing additional studies on these candidates and their status in viral infected cells to add to the manuscript.

We are returning your manuscript with three reviews. The reviewers came to different conclusions about the paper, as you will see. After reading the reviews and looking at the manuscript, we recommend “Major Revision” based on the critiques from the more critical reviews. We am sorry we cannot be more positive at the moment, but we are looking forward to receiving your revision. With extra work and experiments that demonstrate relevance to VZV infection, the manuscript should be suitable for a resubmission, if you so wish to do so. Depending on your revision and the considerations of the comments, we may ask reviewers to reassess the manuscript for the addressing their previous comments and its suitability for publications in PLOS Pathogens upon resubmission. In addition, when you are ready to resubmit, please be prepared to provide the following: (1) A letter containing a detailed list of your responses to the review comments and a description of the changes you have made in the manuscript. (2) Two versions of the manuscript: one with either highlights or tracked changes denoting where the text has been changed; the other a clean version (uploaded as the manuscript file). We hope to receive your revised manuscript within 60 days. If you anticipate any delay in its return, we ask that you let us know the expected resubmission date by replying to this email. Revised manuscripts received beyond 60 days may require evaluation and peer review similar to that applied to newly submitted manuscripts.

We cannot make any decision about publication until we have seen the revised manuscript and your response to the reviewers' comments. Your revised manuscript is also likely to be sent to reviewers for further evaluation.

Sincerely,

Paul Kinchington

Guest Editor

PLOS Pathogens

Erik Flemington

Section Editor

PLOS Pathogens

Kasturi Haldar

Editor-in-Chief

PLOS Pathogens

orcid.org/0000-0001-5065-158X

Michael Malim

Editor-in-Chief

PLOS Pathogens

orcid.org/0000-0002-7699-2064

Reviewer's Responses to Questions

**Part I - Summary**

Reviewer #1: The manuscript by Zhou et al. screens a panel of molecules to learn more about host cell factors that regulate virus spread, which is a hallmark of VZV pathogenesis. The manuscript is well-written and easy to follow. The data are in general of high quality. Cellular molecules that influence VZV spread are an important topic of investigation. In general, the results are new and interesting but the study seems incomplete, and there are problems with data interpretation and conclusions.

Reviewer #2: The experiments described in this manuscript show that calcineurin-phosphatase activity plays a role in regulating VZV-induced cell fusion. This host factor was implicated by the authors' chemical-library screen using cells transfected with the core fusion machinery of the virus. In this experiment, a drug that binds host factor FKBP1A was found to increase cell fusion. That host factor was already known to regulate calcineurin, and the authors nicely show that binding of drug-bound FKBP1A, which blocks phosphatase activity, is needed to increase cell fusion. Thus, some other cellular or viral protein must be de-phosphorylated by calcineurin to keep fusion from getting out of control. The authors also showed that the FKBP1A-binding drug enhanced cell fusion of VZV-infected cells. A variety of other positive and negative inhibitors of fusion were also identified in the screen. Overall, this manuscript makes an important contribution to the literature. Specific comments and concerns are listed below.

1. The experiments that used transfected cells are described in an inaccurate and misleading way. As the authors say in the Methods section, they used a mutant form of gH, which is more fusogenic, but throughout the rest of the text they state that they used "gH", implying that the wild-type was used. To fix this important issue, two changes are needed. First, the rationale for using mutant gH should be stated in the Results section rather than being buried in the Methods. Second, every instance where "gH" is stated in regard to transfections needs to be changed to indicate that the mutant form was used.

2. I could not help but wonder why the authors did not examine any of the inhibitory compounds found in their screens. This is particularly "annoying" being that they argue that enhanced fusion reduces virus spread. Would a drug that decreases cell fusion actually enhance cell spread? I find this omission to be a weakness of the study. If there is a convincing reason for it, that should be included in the manuscript.

3. With regard to the virus experiments (and relating to comment #1), it is important to point out that wild-type virus was used, and hence, the drug-induced enhancement of fusion is not dependent of the mutant form of gH. This is important because once readers realize that a gH mutant was used in the transfection/screening experiment, they will of course wonder whether fusion enhancement occurs with wild-type gH. The authors did not address that in the transfections, but the virus experiments perhaps satisfy the question.

4. By the way, reference #30 is incomplete (but I was able to find and read the paper based on its title).

5. Also with regard to the virus experiments, it would be useful to the naive reader to be told (in the paragraph starting on line #235) that cells within syncytia die to create plaques after further incubation (4 days).

6. The sentence on lines 246-249 is either misleading or data is missing. The authors state that the higher plaque frequencies observed in drug-treated cultures was "attributed" to detachment of cell membrane fragments and reinfection at secondary locations. What is the evidence for that claim? Did the authors include neutralizing antibodies in the medium to show that small satellite plaques were eliminated. If nothing was done along these lines, then the word "attributed" needs to be replaced with "hypothesize".

7. It is not intuitive to a naive reader why enhanced fusion would result in reduced speading or virus production. They might think that enhanced fusion would enable faster spreading and greater opportunity to make more virus. It would be useful to provide a mechanistic explanation or hypothesis in the Discussion. This is not a new observation but it will be confusing to naive readers.

8. The final section of the Results describes mass spectrometry experiments that identified novel substrates for calcineurin. As mentioned above, some protein somewhere in the cell needs to be dephosphorylated to prevent fusion from getting out of control. Here, the authors treated cells with inhibitor and looked for novel proteins that remained phosphorylated. They found five; however, this part of the manuscript is unsatisfying for a few reasons.

8a. The relevance of the five proteins is unclear because only uninfected cells were used. It is abundantly (and nauseatingly) clear that herpesviruses massively alter the physiology of their host cells. Are the five proteins still phosphorylated in drug-treated *infected* cells? Is their expression changed in infected cells? How do we know that these proteins are relevant rather than red herrings?

8b. The authors inappropriately claim on line 276 that they "validated" the five proteins, but they did not. They merely provided further computer-based sequence analyses. To validate their relevance to VZV, they need to actually do experiments!

8c. In the context of an infection, there could be critical *viral* targets for calcineurin. This seems to be ignored by the authors, probably because fusion was enhanced in the gB,gH-gL transfection experiments, but viral infections are vastly more complicated. It would not be surprising to find a viral target. This should be addressed.

8d. A large portion of the Discussion is devoted to these five proteins, but it is unsatisfying because the experiments are incomplete and premature. Would other investigators benefit from the data in its current form? Yes. I do not think it should be eliminated from the manuscript. But it does not deserve so much attention, and the limitations need to be pointed out. All the detailed Discussion can be removed. Any investigator who wishes to pursue these five proteins will have no problem finding the relevant background information.

9. Also in the Discussion, the authors mention previous experiments with HSV-1 showing that a different phosphatase (PTP1B) can regulate cell fusion. My reading of that paper reveals the need for a couple of clarifications here.

9a. Unlike VZV, the HSV-1 fusion machinery in isolation (via transfection) is highly efficient and does not require the use of mutant proteins. Nevertheless, it might be interesting to ask whether inhibition of calcineurin activity would have an effect on HSV-1, and that idea could be added to the Discussion.

9b. Wild-type HSV-1 is not fusogenic. Inhibitors of PTP1B did not enhance fusion. Rather, they prevented the virus from spreading to adjacent cells. I think the HSV-1 study used the artificial situation of cell fusion to get at the mechanism of cell-to-cell spread. Would inhibition of calcineurin activity after HSV-1 spread? That idea could be mentioned in the Discussion.

9c. While the paper showed that HSV-1 is induced to fuse cells when treated with salubrinal (a phenotype that was blocked by inhibitors of PTP1B), the authors showed that the target of salubrinal is *not* eIF2a. Thus, it appears that the target remains unknown.

10. With regard to the Supporting Information, I think S3-Fig is of critical importance and should be included with the main set of figures. Of course, this brings to greater attention the hand-waving on lines 249-250 regarding the "apparent equivalence of VZV growth kinetics", which is also mentioned in comment #6, above.

Reviewer #3: With this manuscript, Arvin, Oliver and colleagues report that inhibition of calcineurin’s phosphatase activity leads to enhanced VZV glycoprotein-mediated cell-cell fusion. They also identify seven host proteins that are phosphorylated at serine or threonine sites if calcineurin’s phosphatase activity is inhibited, suggesting that they might be contributing to regulation of VZV-induced cell-cell fusion, thus securing efficient virus spread.

Overall the data are of good quality, well presented, and appropriately discussed. While the paper does not yet report on studies that start testing if any of those newly identified potential downstream targets of calcineurin’s phosphatase activity is involved in regulating VZV-induced cell-cell fusion, this report of currently available data should still be of interest for others studying virus-induced cell-cell fusion processes. In the following, I thus merely list points that might assist in further improving the manuscript.

Title, Abstract, Introduction, and Discussion

Title and Abstract appropriately summarize the results presented in this paper. The authors also provide a fine introduction into previous investigations of VZV-induced cell fusion and an appropriate, if somewhat short, discussion of the data presented in this report. While this reviewer typically prefers if authors focus most of the discussion on the data that are presented, in this case I think it is appropriate to speculate how one or the other of the newly identified downstream targets of calcineurin may be contributing to regulation of VZV-induced cell fusion.

Minor point: given that their HT-SRFA yields many potential cellular targets that could potentially be factors involved in fusion regulation, it would be interesting to learn why the authors chose to hone in on calcineurin. Was it because of calcineurin’s role in regulating cellular cell-cell fusion processes which is described in the Discussion? If so, I think it would be appropriate to mention this briefly already at the end of the Introduction.

Results

The results section is overall well written, clear, and organized, findings are not overstated.

Minor points, suggestions, comments:

The authors address grouping of fusion enhancers that have a link to neurotransmission. It might be interesting to try their fusion assays using neural cells? (obviously beyond the scope of this paper, but this might provide intriguing additional information, given that previous work of one of the senior authors, PMID 18256143, documented VZV-induced fusion of neurons and satellite cells)

Lines 173-174: when discussing glycoprotein expression levels (total and surface), the authors write that they were “lower in transfected CHO-DSP1 cells treated with pimecrolimus compared to medium or DMSO treatment (S2A Fig),”. It might be useful in this case to have a control stain to show why this might be. Are the cells more resistant to transfection, or does pimecrolimus elicit a larger, non-specific disruption to normal protein production? This would be useful to know as they continue to use the pimecrolimus treatment down the line in the plaque assays and titration curves.

The authors nicely differentiate between the various potential mechanisms of FKBP1A (impacting calcineurin vs impacting mTOR using specific inhibitors with known function), and validate the impact on calcineurin by multiple approaches.

Line 204: the authors mention a “random shRNA control”, but unless I miss it, there is no description of that sequence in the methods, nor do they describe the generation of such a construct?

Lines 218-221: “These findings corroborated the NFATC1 nuclear translocation data, indicating that FKBP1A was the specific cellular protein bound by pimecrolimus to produce the inhibitory effect on calcineurin and confirmed that dysregulation of VZV gB/gH-gL mediated cell fusion by pimecrolimus was a result from the inhibition of calcineurin phosphatase activity”. I can see how the authors are connecting these dots, but I think they should complete experiments to at least investigate some metric of calcineurin activity within the context of an actual VZV infection rather than in uninfected cells, or they should soften their statement here (specifically as they use the phrasing “confirmed that dysregulation of VZV”).

Figure 3B labelling suggests that nuclei were labeled with DAPI, yet the legend says Hoechst. A quick search shows no mention of DAPI in the text, assuming this was a typo?

Figure 3B shows difference between localization within the KD or control, but doesn’t show how baseline localization may vary between KD and control which would be interesting to know. The way the data was normalized makes it impossible to know. Also, since the data in 3B are from two replicates, there should not be error bars and statistical analysis. Rather, the individual replicates should be shown. Same goes for C where there are only 3 replicates.

The discussion of S3Fig A and B is out of order. Perhaps one could reorganize the figure or the discussion of the results so that the two pieces are more reflective of each other?

Fig. 4C: please show the individual data points. Ideally, avoid statistical testing, but if needed (i.e. if the result is not abundantly clear in the dot plot), it should at least be nonparametric unless normal distribution can be demonstrated.

Fig. 5B: not sure if it is appropriate here to take each plaque as a data point, there really are only two replicates here, hence statistics and error bars may not be appropriate?

Materials and Methods

Lines 397-401 describe that the authors used a CTD-mutant gH protein for their screening protocol (and presumably also for some of the subsequent assays?). However, as mentioned in the Introduction, the CTD of gB and gH affects fusion regulation and even cellular expression profiles. While in subsequent experiments the effects on fusion and plaque formation are shown with WT gH-expressing virus, thus validating them, usage of mutant gH should nevertheless be mentioned in the Results section.

**Part II – Major Issues: Key Experiments Required for Acceptance**

Reviewer #1: 1)If the authors would like to make this study about calcineurin, then its role in VZV fusion should be directly tested, e.g. by knockdown. The conclusions about calcineurin and VZV are overstated because the approaches are quite indirect. The inhibitors in the study, pimecrolimus and related molecules, target FKBP1A, not calcineurin itself. One could also argue that the big picture story here is more about calcium flux and less about calcineurin or FKBP1A.

2)The identification of host cell substrates for calcineurin in MeWo cells is interesting from a cell biology perspective, but there is no evidence that these substrates have anything to do with VZV infection. The paper does not test the role of these substrates in VZV fusion and spread.

3)The terms fusion and spread as used here are a bit misleading and confusing. A larger plaque size means the plaque encompasses a larger area or volume, but does not mean spread to more cells. The results presented exemplify this. The number of nuclei that comprise a syncytium is a more accurate reflection of both the ability of a transfected/infected cell to fuse with AND spread to other cells in the culture.

Inhibition of calcineurin phosphatase activity enhanced VZV-induced cell fusion but reduced VZV spread. These disparate results are not clearly addressed. If there is enhancement in one assay and decrease in another, then the assays are not measuring the same thing. We are left without a sense of what a decrease in syncytium size really means (if it is not related to a decrease in fusion) and why it is relevant to VZV pathogenesis.

Reviewer #2: None

Reviewer #3: n/a

**Part III – Minor Issues: Editorial and Data Presentation Modifications**

Reviewer #1: The manuscript assesses chemical inhibitors and knockdown of FKBP1A on VZV cell fusion and plaque size, yet the background on FKBP1A and its role in the cell, relationship to calcineurin, and calcium is buried. There should be more focus on FKBP1A in the writing. It should be better described in the abstract and introduction.

Reviewer #2: None beyond what was mentioned in the review

Reviewer #3: included in Part I

PLOS authors have the option to publish the peer review history of their article (what does this mean?). If published, this will include your full peer review and any attached files.

Reviewer #1: No

Reviewer #2: No

Reviewer #3: No
---

## [Editor Report · Decision Letter 1]

2 Oct 2020

Dear Dr Zhou

I have carefully read your revised manuscript and your responses to the three reviewers of the initial submission. I have come to the conclusion that the revised version of the manuscript has more than adequately addressed the concerns raised about the initial submission, and that it does not require further external review.

As such, we are pleased to inform you that your manuscript 'Calcineurin phosphatase activity regulates Varicella-Zoster Virus induced cell-cell fusion' has been provisionally accepted for publication in PLOS Pathogens.

Best regards,

Paul Kinchington

Guest Editor

PLOS Pathogens

Erik Flemington

Section Editor

PLOS Pathogens

Kasturi Haldar

Editor-in-Chief

PLOS Pathogens

orcid.org/0000-0001-5065-158X

Michael Malim

Editor-in-Chief

PLOS Pathogens

orcid.org/0000-0002-7699-2064
---

## [Editor Report · Acceptance letter]

27 Oct 2020

Dear Dr Zhou,

We are delighted to inform you that your manuscript, "Calcineurin phosphatase activity regulates Varicella-Zoster Virus induced cell-cell fusion," has been formally accepted for publication in PLOS Pathogens.

Best regards,

Kasturi Haldar

Editor-in-Chief

PLOS Pathogens

orcid.org/0000-0001-5065-158X

Michael Malim

Editor-in-Chief

PLOS Pathogens

orcid.org/0000-0002-7699-2064